# ACCELERATED SINGLE-CALL METHODS FOR CONSTRAINED MIN-MAX OPTIMIZATION

**Yang Cai**
Yale University
yang.cai@yale.edu

**Weiqiang Zheng**
Yale University
weiqiang.zheng@yale.edu

## ABSTRACT

We study first-order methods for constrained min-max optimization. Existing methods either require two gradient calls or two projections in each iteration, which may be costly in some applications. In this paper, we first show that a variant of the *Optimistic Gradient (OG)* method, a *single-call single-projection* algorithm, has $O(\frac{1}{\sqrt{T}})$ best-iterate convergence rate for inclusion problems with operators that satisfy the weak Minty variation inequality (MVI). Our second result is the first single-call single-projection algorithm – the *Accelerated Reflected Gradient (ARG)* method that achieves the *optimal* $O(\frac{1}{T})$ last-iterate convergence rate for inclusion problems that satisfy negative comonotonicity. Both the weak MVI and negative comonotonicity are well-studied assumptions and capture a rich set of non-convex non-concave min-max optimization problems. Finally, we show that the *Reflected Gradient (RG)* method, another *single-call single-projection* algorithm, has $O(\frac{1}{\sqrt{T}})$ last-iterate convergence rate for constrained convex-concave min-max optimization, answering an open problem of (Hsieh et al., 2019). Our convergence rates hold for standard measures such as the tangent residual and the natural residual.

## 1  INTRODUCTION

Various Machine Learning applications, from the generative adversarial networks (GANs) (e.g., (Goodfellow et al., 2014; Arjovsky et al., 2017)), adversarial examples (e.g., (Madry et al., 2017)), robust optimization (e.g., (Ben-Tal et al., 2009)), to reinforcement learning (e.g., (Du et al., 2017; Dai et al., 2018)), can be captured by constrained min-max optimization. Unlike the well-behaved convex-concave setting, these modern ML applications often require solving non-convex non-concave min-max optimization problems in high dimensional spaces.

Unfortunately, the general non-convex non-concave setting is intractable even for computing a *local* solution (Hirsch et al., 1989; Papadimitriou, 1994; Daskalakis et al., 2021). Motivated by the intractability, researchers turn their attention to non-convex non-concave settings *with structure*. Significant progress has been made for several interesting structured non-convex non-concave settings, such as the ones that satisfy the weak Minty variation inequality (MVI) (Definition 2) (Diakonikolas et al., 2021; Pethick et al., 2022) and the ones that satisfy the more strict negatively comonotone condition (Definition 3) (Lee & Kim, 2021a; Cai et al., 2022a). These algorithms are variations of the celebrated extragradient (EG) method (Korpelevich, 1976), an iterative first-order method. Similar to the extragradient method, these algorithms all require *two oracle calls* per iteration, which may be costly in practice. We investigate the following important question in this paper:

> *Can we design efficient **single-call** first-order methods for*
>
> *structured non-convex non-concave min-max optimization?*        (*)

We provide an affirmative answer to the question. We first show that a single-call method known as the *Optimistic Gradient* (OG) method (Hsieh et al., 2019) is applicable to all non-

convex non-concave settings that satisfy the *weak MVI*. We then provide the *Accelerated Reflected Gradient (ARG)* method that achieves the optimal convergence rate in all non-convex non-concave settings that satisfy the *negatively comonotone condition*. Single-call methods have been studied in the convex-concave settings (Hsieh et al., 2019) but not for the more general non-convex non-concave settings. See Table 1 for comparisons between our algorithms and other algorithms from the literature.

| | Algorithm | 1-Call? | Constraints? | Non-Monotone Comonotone | weak MVI |
|---|---|---|---|---|---|
| Normal | EG+ (Diakonikolas et al., 2021) | ✗ | ✗ | $O(\frac{1}{\sqrt{T}})$ | $O(\frac{1}{\sqrt{T}})$ |
| | CEG+ (Pethick et al., 2022) | ✗ | ✓ | $O(\frac{1}{\sqrt{T}})$ | $O(\frac{1}{\sqrt{T}})$ |
| | OGDA (Böhm, 2022; Bot et al., 2022) | ✓ | ✗ | $O(\frac{1}{\sqrt{T}})$ | $O(\frac{1}{\sqrt{T}})$ |
| | OG[**this paper**] | ✓ | ✓ | $O(\frac{1}{\sqrt{T}})$ | $O(\frac{1}{\sqrt{T}})$ |
| Accelerated | FEG (Lee & Kim, 2021b) | ✗ | ✗ | $O(\frac{1}{T})$ | |
| | AS (Cai et al., 2022a) | ✗ | ✓ | $O(\frac{1}{T})$ | |
| | ARG [**This paper**] | ✓ | ✓ | $O(\frac{1}{T})$ | |

Table 1: Existing results for min-max optimization problem with non-monotone operators. A ✓ in "Constraints?" means the algorithm works in the constrained setting. The convergence rate is in terms of the operator norm (in the unconstrained setting) and the residual (in the constrained setting).

## 1.1 OUR CONTRIBUTIONS

Throughout the paper, we adopt the more general and abstract framework of inclusion problems, which includes constrained min-max optimization as a special case. More specifically, we consider the following problem.

**Inclusion Problem.** Given $E = F + A$ where $F : \mathbb{R}^n \to \mathbb{R}^n$ is a single-valued (possibly non-monotone) operator and $A : \mathbb{R}^n \rightrightarrows \mathbb{R}^n$ is a set-valued maximally monotone operator, the *inclusion* problem is defined as follows

$$\text{find } z^* \in \mathcal{Z} \text{ such that } \mathbf{0} \in E(z^*) = F(z^*) + A(z^*). \tag{IP}$$

As shown in the following example, we can interpret a min-max optimization problem as an inclusion problem.

**Example 1** (Min-Max Optimization). *The following structured min-max optimization problem captures a wide range of applications in machine learning such as GANs, adversarial examples, robust optimization, and reinforcement learning:*

$$\min_{x \in \mathbb{R}^{n_x}} \max_{y \in \mathbb{R}^{n_y}} f(x, y) + g(x) - h(y), \tag{1}$$

*where $f(\cdot, \cdot)$ is possibly non-convex in $x$ and non-concave in $y$. Regularized and constrained min-max problems are covered by appropriate choices of lower semi-continuous and convex functions $g$ and $h$. Examples include the $\ell_1$-norm, the $\ell_2$-norm, and the indicator function of a closed convex feasible set. Let $z = (x, y)$, if we define $F(z) = (\partial_x f(x, y), -\partial_y f(x, y))$ and $A(z) = (\partial g(x), \partial h(y))$, where $A$ is maximally monotone, then the first-order optimality condition of (1) has the form of an inclusion problem.*

(Daskalakis et al., 2021) shows that without any assumption on the operator $E = F + A$, the problem is intractable.[1] The most well understood setting is when $E$ is monotone, i.e., $\langle u - v, z - z' \rangle \geq 0$ for all $z, z'$ and $u \in E(z)$, $v \in E(z')$, which captures convex-concave min-max optimization. Motivated by non-convex non-concave min-max optimization, we consider the two most widely studied families of non-monotone operators: (i) negatively

---

[1]Indeed, even if $A$ is maximally monotone, (Daskalakis et al., 2021) implies that the problem is still intractable without further assumptions on $F$.

comonotone operators and (ii) operators that satisfy the less restrictive weak MVI. See Section 2 for more detailed discussion on their relationship. Here are the main contributions of this paper.

**Contribution 1:** We provide an extension of the *Optimistic Gradient (OG)* method for inclusion problems when the operator $E = F + A$ satisfies the weak MVI. More specifically, we prove that OG has a $O(\frac{1}{\sqrt{T}})$ convergence rate (Theorem 1) matching the state of the art algorithms (Diakonikolas et al., 2021; Pethick et al., 2022). Importantly, our algorithm only requires a single oracle call to $F$ and a single call to the resolvent of $A$.[a]

---
[a] The resolvent of A is defined as $(I + A)^{-1}$. When $A$ is the subdifferential of the indicator function of a closed convex set, the resolvent operator is exactly the Euclidean projection. Hence our algorithm performs a single projection in the constrained case.

Next, we provide an accelerated single-call method when the operator satisfies the stronger negatively comonotone condition.

**Contribution 2:** We design an accelerated version of the *Reflected Gradient (RG)* (Chambolle & Pock, 2011; Malitsky, 2015; Cui & Shanbhag, 2016; Hsieh et al., 2019) method that we call the *Accelerated Reflected Gradient* (ARG) method, which has the optimal $O(\frac{1}{T})$ convergence rate for inclusion problems whose operators $E = F + A$ are negatively comonotone (Theorem 2). Note that $O(\frac{1}{T})$ is the optimal convergence rate for any first-order methods even for monotone inclusion problems (Diakonikolas, 2020; Yoon & Ryu, 2021). Importantly, ARG only requires a single oracle call to $F$ and a single call to the resolvent of $A$.

Finally, we resolve an open question from (Hsieh et al., 2019).

**Contribution 3:** We show that the *Reflected Gradient (RG)* method has a *last-iterate* convergence rate of $O(\frac{1}{\sqrt{T}})$ for constrained convex-concave min-max optimization (Theorem 3). Hsieh et al. (2019) show that the RG algorithm asymptotically converges but fails to obtain a concrete rate. We strengthen their result to obtain a tight finite convergence rate for RG.

We also provide illustrative numerical experiments in Appendix E.

## 1.2 RELATED WORKS

We provide a brief discussion of the most relevant and recent results on nonconvex-nonconcave min-max optimization here and defer the discussion on related results in the convex-concave setting to Appendix A. We also refer readers to (Facchinei & Pang, 2003; Bauschke & Combettes, 2011; Ryu & Yin, 2022) and references therein for a comprehensive literature review on inclusion problems and related variational inequality problems.

**Structured Nonconvex-Nonconcave Min-Max Optimization.** Since in general nonconvex-nonconcave min-max optimization problems are intractable, recent works study problems under additional assumptions. The *Minty variational inequality* (MVI) assumption (also called coherence or variational stability), which covers all quasiconvex-concave and starconvex-concave problems, is well-studied in e.g., (Dang & Lan, 2015; Zhou et al., 2017; Liu et al., 2019; Malitsky, 2020; Song et al., 2020; Liu et al., 2021). Extragradient-type algorithms has $O(\frac{1}{\sqrt{T}})$ convergence rate for problems that satisfies MVI (Dang & Lan, 2015).

Diakonikolas et al. (2021) proposes a weaker assumption called *weak MVI*, which includes both MVI or negative comonotonicity (Bauschke et al., 2021) as special cases. Under the weak MVI, the EG+ (Diakonikolas et al., 2021) and OGDA+ (Böhm, 2022) algorithms have $O(\frac{1}{\sqrt{T}})$ convergence rate in the unconstrained setting. Recently, Pethick et al. (2022) gen-

eralizes EG+ to CEG+ algorithm, achieving the same convergence rate in the general (constrained) setting. To the best of our knowledge, the OG algorithm is the only single-call single-resolvent algorithm with $O(\frac{1}{\sqrt{T}})$ convergence rate when we only assume weak MVI (Theorem 1).

The result for accelerated algorithms in the nonconvex-nonconcave setting is sparser. For negatively comonotone operators, optimal $O(\frac{1}{T})$ convergence rate is achieved by variants of the EG algorithm in the unconstrained setting (Lee & Kim, 2021a) and in the constrained setting (Cai et al., 2022a) . To the best of our knowledge, the ARG algorithm is the first efficient single-call single-resolvent method that achieves the accelerated and optimal $O(\frac{1}{T})$ convergence rate in the constrained nonconvex-nonconcave setting (Theorem 2). We summarize previous results and our results in Table 1. Our analysis of ARG is inspired by (Cai et al., 2022a) and uses a similar potential function argument.

## 2 PRELIMINARIES

**Basic Notations.** Throughout the paper, we focus on the Euclidean space $\mathbb{R}^n$ equipped with inner product $\langle \cdot, \cdot \rangle$. We denote the standard $\ell_2$-norm by $\| \cdot \|$. For any closed and convex set $\mathcal{Z} \subseteq \mathbb{R}^n$, $\Pi_{\mathcal{Z}}[\cdot] : \mathbb{R}^n \to \mathcal{Z}$ denotes the Euclidean projection onto set $\mathcal{Z}$ such that for any $z \in \mathbb{R}^n$, $\Pi_{\mathcal{Z}}[z] = \operatorname{argmin}_{z' \in \mathcal{Z}} \|z - z'\|$. We denote $\mathcal{B}(z, r)$ the $\ell_2$-ball centered at $z$ with radius $r$.

**Normal Cone.** We denote $N_{\mathcal{Z}} : \mathcal{Z} \to \mathbb{R}^n$ to be the normal cone operator such that for $z \in \mathcal{Z}$, $N_{\mathcal{Z}}(z) = \{a : \langle a, z' - z \rangle \leq 0, \forall z' \in \mathcal{Z}\}$. Define the indicator function

$$\mathbb{I}_{\mathcal{Z}}(z) = \begin{cases} 0 & \text{if } z \in \mathcal{Z}, \\ +\infty & \text{otherwise.} \end{cases}$$

It is not hard to see that the subdifferential operator $\partial \mathbb{I}_{\mathcal{Z}} = N_{\mathcal{Z}}$. A useful fact is that if $z = \Pi_{\mathcal{Z}}[z']$, then $\lambda(z' - z) \in N_{\mathcal{Z}}(z)$ for any $\lambda \geq 0$.

**Monotone Operator.** We recall some standard definitions and results on monotone operators here and refer the readers to (Bauschke & Combettes, 2011; Ryu & Boyd, 2016; Ryu & Yin, 2022) for more detailed introduction. A *set-valued operator* $A : \mathbb{R}^n \rightrightarrows \mathbb{R}^n$ maps each point $z \in \mathbb{R}^n$ to a subset $A(z) \subseteq \mathbb{R}^n$. We denote the *graph* of $A$ as $\operatorname{Gra}(A) := \{(z, u) : u \in A(z)\}$ and the *zeros* of $A$ as $\operatorname{Zer}(A) = \{z : \mathbf{0} \in A(z)\}$. The inverse operator of $A$ is denoted as $A^{-1}$ whose graph is $\operatorname{Gra}(A^{-1}) = \{(u, z) : (z, u) \in \operatorname{Gra}(A)\}$. For two operators $A$ and $B$, we denote $A + B$ to be the operator with graph $\operatorname{Gra}(A + B) = \{(z, u_A + u_B) : (z, u_A) \in \operatorname{Gra}(A), (z, u_B) \in \operatorname{Gra}(B)\}$. We denote the identity operator as $I : \mathbb{R}^n \to \mathbb{R}^n$. We say operator $A$ is *single valued* if $|A(z)| \leq 1$ for all $z \in \mathbb{R}^n$. Single-valued operator $A$ is *L-Lipschitz* if $\|A(z) - A(z')\| \leq L \cdot \|z - z'\|, \forall z, z' \in \mathbb{R}^n$. Moreover, we say $A$ is *non-expansive* if it is 1-Lipschitz.

**Definition 1** ((Maximally) monotonicity)**.** *An operator $A : \mathbb{R}^n \rightrightarrows \mathbb{R}^n$ is* monotone *if*

$$\langle u - u', z - z' \rangle \geq 0, \quad \forall (z, u), (z', u') \in \operatorname{Gra}(A).$$

*Moreover, $A$ is* maximally monotone *if $A$ is monotone and $\operatorname{Gra}(A)$ is not properly contained in the graph of any other monotone operators.*

When $g : \mathbb{R}^n \to \mathbb{R}^n$ is closed, convex, and proper, then its subdifferential operator $\partial g$ is maximally monotone. As an example, the normal cone operator $N_{\mathcal{Z}} = \partial \mathbb{I}_{\mathcal{Z}}$ is maximally monotone.

We denote **resolvent** of $A$ as $J_A = (I + A)^{-1}$. Some useful properties of the resolvent are summarized in the following proposition.

**Proposition 1.** *If $A$ is maximally monotone, then $J_A$ satisfies the following.*

*1. The domain of $J_A$ is $\mathbb{R}^n$. $J_A$ is non-expansive and single-valued on $\mathbb{R}^n$.*

2. *If $z = J_A(z')$, then $z' - z \in A(z)$. If $c \in A(z)$, then $z = J_A(z + c)$.*

3. *When $A = N_{\mathcal{Z}}$ is the normal cone operator for some closed convex set $\mathcal{Z}$, then $J_{\eta A} = \Pi_{\mathcal{Z}}$ is the Euclidean projection onto $\mathcal{Z}$ for all $\eta > 0$.*

**Non-Monotone Operator.**

**Definition 2** (Weak MVI (Diakonikolas et al., 2021; Pethick et al., 2022)). *An operator $A : \mathbb{R}^n \rightrightarrows \mathbb{R}^n$ satisfies* weak MVI *if for some $z^* \in \text{Zer}(A)$, there exists $\rho \le 0$*

$$\langle u, z - z^* \rangle \ge \rho \|u\|^2, \quad \forall (z, u) \in \text{Gra}(A).$$

**Definition 3** (Comonotonicity (Bauschke et al., 2021)). *An operator $A : \mathbb{R}^n \rightrightarrows \mathbb{R}^n$ is $\rho$-comonotone if*

$$\langle u - u', z - z' \rangle \ge \rho \|u - u'\|^2, \quad \forall (z, u), (z', u') \in \text{Gra}(A).$$

When $A$ is $\rho$-comonotone for $\rho > 0$, then $A$ is also known as $\rho$-cocoercive, which is a stronger condition than monotonicity. When $A$ is $\rho$-comonotone for $\rho < 0$, then $A$ is non-monotone. Weak MVI with $\rho = 0$ is also know as MVI, coherence, or variational stability. Note that the weak MVI is implied by negative comonotonicity. We refer the readers to (Lee & Kim, 2021a, Example 1), (Diakonikolas et al., 2021, Section 2.2) and (Pethick et al., 2022, Section 5) for examples of min-max optimization problems that satisfy the two conditions.

## 2.1 PROBLEM FORMULATION

**Inclusion Problem.** Given $E = F + A$ where $F : \mathbb{R}^n \to \mathbb{R}^n$ is a single-valued (possibly non-monotone) operator and $A : \mathbb{R}^n \rightrightarrows \mathbb{R}^n$ is a set-valued maximally monotone operator, the *inclusion* problem is defined as follows

$$\text{find } z^* \in \mathcal{Z} \text{ such that } \mathbf{0} \in E(z^*) = F(z^*) + A(z^*). \tag{IP}$$

We say $z$ is an $\epsilon$-approximate solution to an inclusion problem (IP) if $\mathbf{0} \in F(z) + A(z) + \mathcal{B}(\mathbf{0}, \epsilon)$. Throughout the paper, we study IP problems under the following assumption.

**Assumption 1.** *In the setup of IP,*

1. *there exists $z^* \in \text{Zer}(E)$, i.e., $\mathbf{0} \in F(z^*) + A(z^*)$.*

2. *$F$ is L-Lipschitz.*

3. *$A$ is maximally monotone.*

When $F$ is monotone, we refer to the corresponding IP problem as a *monotone inclusion* problem, which covers convex-concave min-max optimization. In the more general non-monotone setting, we would study problems that satisfy negative comonotonicity or weak MVI.

**Assumption 2** (Comonotonicity). *In the setup of IP, $E = F + A$ is $\rho$-comonotone, i.e.,*

$$\langle u - u', z - z' \rangle \ge \rho \|u - u'\|^2, \quad \forall (z, u), (z', u') \in \text{Gra}(E).$$

**Assumption 3** (Weak MVI). *In the setup of IP, $E = F + A$ satisfies weak MVI with $\rho \le 0$, i.e., there exists $z^* \in \text{Zer}(E)$,*

$$\langle u, z - z^* \rangle \ge \rho \|u\|^2, \quad \forall (z, u) \in \text{Gra}(E).$$

An important special case of inclusion problem is the variational inequality problem.

**Variational Inequality.** Let $\mathcal{Z} \subseteq \mathbb{R}^n$ be a closed and convex set and $F : \mathbb{R}^n \to \mathbb{R}^n$ be a single-valued operator. The *variation inequality* (VI) problem associated with $\mathcal{Z}$ and $F$ is stated as

$$\text{find } z^* \in \mathcal{Z} \text{ such that } \langle F(z^*), z^* - z \rangle \le 0, \forall z \in \mathcal{Z}. \tag{VI}$$

Note that VI is a special case of IP when $A = N_{\mathcal{Z}} = \partial \mathbb{I}_{\mathcal{Z}}$ is the normal cone operator:

$$0 \in F(z^*) + N_{\mathcal{Z}}(z^*) \Leftrightarrow -F(z^*) \in N_{\mathcal{Z}}(z^*) \Leftrightarrow \langle F(z^*), z^* - z \rangle \leq 0, \forall z \in \mathcal{Z}.$$

The general formulation of VI unifies many problems such as convex optimization, min-max optimization, computing Nash equilibria in multi-player concave games, and is extensively-studied since 1960s (Facchinei & Pang, 2003). Definitions of the convergence measure for VI and the classical algorithms, EG and PEG, are presented in Appendix B.

## 2.2 CONVERGENCE MEASURE

We focus on a strong convergence measure called the *tangent residual*, defined as $r_{F,A}^{tan}(z) := \min_{c \in A(z)} \|F(z) + c\|$. It is clear by definition that $r_{F,A}^{tan}(z) \leq \epsilon$ implies $z$ is an $\epsilon$-approximate solution to the inclusion (IP) problem, and also an $(\epsilon \cdot D)$ approximate strong solution to the corresponding variational inequality (VI) problem when $\mathcal{Z}$ is bounded by $D$. Moreover, the tangent residual is an upper bound of other notion of residuals in the literature such as the natural residual $r_{F,A}^{nat}$ (Diakonikolas, 2020) or the forward-backward residual $r_{F,A}^{fb}$ (Yoon & Ryu, 2022) as shown in Proposition 2 (see Appendix B.4 for the formal statement and proof). Thus our convergence rates on the tangent residual also hold for the natural residual or the forward-backward residual. Note that in the unconstrained setting where $A = 0$, these residuals are all equivalent to the *operator norm* $\|F(z)\|$.

## 3 OPTIMISTIC GRADIENT METHOD FOR WEAK MVI PROBLEMS

In this section, we consider an extension of the *Optimistic Gradient* (OG) algorithm (Daskalakis et al., 2017; Mokhtari et al., 2020a;b; Hsieh et al., 2019; Peng et al., 2020) for inclusion problems: given arbitrary starting point $z_{-\frac{1}{2}} = z_0 \in \mathbb{R}^n$ and step size $\eta > 0$, the update rule is

$$\begin{aligned} z_{t+\frac{1}{2}} &= J_{\eta A}\left[z_t - \eta F(z_{t-\frac{1}{2}})\right], \\ z_{t+1} &= z_{t+\frac{1}{2}} + \eta F(z_{t-\frac{1}{2}}) - \eta F(z_{t+\frac{1}{2}}). \end{aligned} \tag{OG}$$

For $t \geq 1$, the update rule can also be written as $z_{t+\frac{3}{2}} = J_{\eta A}[z_{t+\frac{1}{2}} - 2\eta F(z_{t+\frac{1}{2}}) + \eta F(z_{t-\frac{1}{2}})]$, which coincides with the forward-reflected-backward algorithm (Malitsky & Tam, 2020). We remark that the update rule of OG is different from the *Optimistic Gradient Descent/Ascent (OGDA)* algorithm (also known as *Past Extra Gradient (PEG)* algorithm) (Popov, 1980), which is single-call but requires two projections in each iteration.

Previous results for OG only hold in the convex-concave (monotone) setting. The main result in this section is that OG has $O(\frac{1}{\sqrt{T}})$ convergence rate even for nonconvex-nonconcave min-max optimization problems that satisfy weak MVI, matching the state of the art results achieved by two-call methods (Diakonikolas et al., 2021; Pethick et al., 2022). Remarkably, OG only requires single call to $F$ and single call to the resolvent $J_{\eta A}$ in each iteration. The main result is shown in Theorem 1. The proof relies on a simple yet important observation that $\frac{z_t - z_{t+1}}{\eta} \in F(z_{t+\frac{1}{2}}) + A(z_{t+\frac{1}{2}})$.

**Theorem 1.** *Assume Assumption 1 and 3 hold with $\rho \in (-\frac{1}{12\sqrt{3}L}, 0]$. Consider the iterates of* (OG) *with step size $\eta \in (0, \frac{1}{2L})$ satisfying $C = \frac{1}{2} + \frac{2\rho}{\eta} - 2\eta^2 L^2 > 0$ (existence of such $\eta$ is guaranteed by Fact 1). Then for any $T \geq 1$,*

$$\min_{t \in [T]} r_{F,A}^{tan}(z_{t+\frac{1}{2}})^2 \leq \min_{t \in [T]} \frac{\|z_{t+1} - z_t\|^2}{\eta^2} \leq \frac{H^2}{C\eta^2} \cdot \frac{1}{T},$$

*where $H^2 = \|z_1 - z^*\|^2 + \frac{1}{4}\|z_{\frac{1}{2}} - z_0\|^2$.*

*Proof.* From the update rule of (OG), we have the following identity (see also (Hsieh et al., 2019, Appendix B)): for any $p \in \mathcal{Z}$,

$$\|z_{t+1} - p\|^2 = \|z_t - p\|^2 + \left\|z_{t+1} - z_{t+\frac{1}{2}}\right\|^2 - \left\|z_{t+\frac{1}{2}} - z_t\right\|^2$$
$$+ 2\left\langle z_t - \eta F(z_{t-\frac{1}{2}}) - z_{t+\frac{1}{2}} + \eta F(z_{t+\frac{1}{2}}), p - z_{t+\frac{1}{2}}\right\rangle. \tag{2}$$

Since $z_{t+\frac{1}{2}} = J_{\eta A}[z_t - \eta F(z_{t-\frac{1}{2}})]$, we have $\frac{z_t - \eta F(z_{t-\frac{1}{2}}) - z_{t+\frac{1}{2}}}{\eta} \in A(z_{t+\frac{1}{2}})$ by Proposition 1. Then

$$\frac{z_t - z_{t+1}}{\eta} = \frac{z_t - \eta F(z_{t-\frac{1}{2}}) - z_{t+\frac{1}{2}}}{\eta} + F(z_{t+\frac{1}{2}}) \in F(z_{t+\frac{1}{2}}) + A(z_{t+\frac{1}{2}}).$$

Set $p = z^*$. By the weak MVI assumption, we have

$$2\left\langle z_t - \eta F(z_{t-\frac{1}{2}}) - z_{t+\frac{1}{2}} + \eta F(z_{t+\frac{1}{2}}), z^* - z_{t+\frac{1}{2}}\right\rangle = 2\eta\left\langle \frac{z_t - z_{t+1}}{\eta}, z^* - z_{t+\frac{1}{2}}\right\rangle$$
$$\leq -\frac{2\rho}{\eta}\|z_t - z_{t+1}\|^2. \tag{3}$$

Define $c = \frac{1}{2} - 2\eta^2 L^2 > 0$. We have identity

$$(1 - 2c)\eta^2 L^2 = 4\eta^4 L^4 = \frac{1}{2} - c - (1 + 2c)\eta^2 L^2. \tag{4}$$

Combining Equation (2) and (3) and using $\|a + b\|^2 \leq 2\|a\|^2 + 2\|b\|^2$, we have

$$\|z_{t+1} - z^*\|^2$$
$$\leq \|z_t - z^*\|^2 + \left\|z_{t+1} - z_{t+\frac{1}{2}}\right\|^2 - \left\|z_{t+\frac{1}{2}} - z_t\right\|^2 + c\|z_t - z_{t+1}\|^2 - (c + \frac{2\rho}{\eta})\|z_t - z_{t+1}\|^2$$
$$\leq \|z_t - z^*\|^2 + (1 + 2c)\left\|z_{t+1} - z_{t+\frac{1}{2}}\right\|^2 - (1 - 2c)\left\|z_{t+\frac{1}{2}} - z_t\right\|^2 - (c + \frac{2\rho}{\eta})\|z_t - z_{t+1}\|^2. \tag{5}$$

Using the update rule of OG and $L$-Lipschitzness of $F$, we have that for any $t \geq 0$,

$$\left\|z_{t+1} - z_{t+\frac{1}{2}}\right\|^2 = \left\|\eta F(z_{t-\frac{1}{2}}) - \eta F(z_{t+\frac{1}{2}})\right\|^2 \leq \eta^2 L^2 \left\|z_{t+\frac{1}{2}} - z_{t-\frac{1}{2}}\right\|^2. \tag{6}$$

Moreover, using $\|a + b\|^2 \leq 2\|a\|^2 + 2\|b\|^2$ and Equation (6), we have that for any $t \geq 1$,

$$\left\|z_{t+\frac{1}{2}} - z_{t-\frac{1}{2}}\right\|^2 \leq 2\left\|z_{t+\frac{1}{2}} - z_t\right\|^2 + 2\left\|z_t - z_{t-\frac{1}{2}}\right\|^2 \leq 2\left\|z_{t+\frac{1}{2}} - z_t\right\|^2 + 2\eta^2 L^2\left\|z_{t-\frac{1}{2}} - z_{t-\frac{3}{2}}\right\|^2.$$

which imples

$$\left\|z_{t+\frac{1}{2}} - z_t\right\|^2 \geq \frac{1}{2}\left\|z_{t+\frac{1}{2}} - z_{t-\frac{1}{2}}\right\|^2 - \eta^2 L^2\left\|z_{t-\frac{1}{2}} - z_{t-\frac{3}{2}}\right\|^2. \tag{7}$$

Combining Equation (4), (5), (6), and (7), we have that for all $t \geq 1$.

$$\|z_{t+1} - z^*\|^2$$
$$\leq \|z_t - z^*\|^2 + (1 + 2c)\left\|z_{t+1} - z_{t+\frac{1}{2}}\right\|^2 - (1 - 2c)\left\|z_{t+\frac{1}{2}} - z_t\right\|^2 - (c + \frac{2\rho}{\eta})\|z_t - z_{t+1}\|^2$$
$$\leq \|z_t - z^*\|^2 + (1 - 2c)\eta^2 L^2\left\|z_{t-\frac{1}{2}} - z_{t-\frac{3}{2}}\right\|^2 - \left(\frac{1}{2} - c - (1 + 2c)\eta^2 L^2\right)\left\|z_{t+\frac{1}{2}} - z_{t-\frac{1}{2}}\right\|^2$$
$$- (c + \frac{2\rho}{\eta})\|z_t - z_{t+1}\|^2$$
$$= \|z_t - z^*\|^2 + 4\eta^4 L^4\left(\left\|z_{t-\frac{1}{2}} - z_{t-\frac{3}{2}}\right\|^2 - \left\|z_{t+\frac{1}{2}} - z_{t-\frac{1}{2}}\right\|^2\right) - (c + \frac{2\rho}{\eta})\|z_t - z_{t+1}\|^2.$$

Telescoping the above inequality and using $c = \frac{1}{2} - 2\eta^2 L^2$ and $\eta L < \frac{1}{2}$, we get

$$(\frac{1}{2} + \frac{2\rho}{\eta} - 2\eta^2 L^2) \sum_{t=1}^{T} \|z_t - z_{t+1}\|^2 \leq \|z_1 - z^*\|^2 + \frac{1}{4}\left\|z_{\frac{1}{2}} - z_{-\frac{1}{2}}\right\|^2.$$

Note that $z_0$ is the same as $z_{-\frac{1}{2}}$. This completes the proof. □

**Fact 1.** *For any $L > 0$ and $\rho > -\frac{1}{12\sqrt{3}L}$. There exists $\eta \in (0, \frac{1}{2L})$ such that $\frac{1}{2} + \frac{2\rho}{\eta} - 2\eta^2 L^2 > 0$.*

*Proof.* Let $\eta = \frac{1}{2\sqrt{3}L}$, then the desired inequality holds whenever

$$\rho > \frac{\eta L(1 - 4\eta^2 L^2)}{4} \cdot \frac{1}{L} = -\frac{1}{12\sqrt{3}L}.$$

□

# 4 ACCELERATED REFLECTED GRADIENT FOR NEGATIVELY COMONOTONE PROBLEMS

In this section, we propose a new algorithm called the *Accelerated Reflected Gradient* (ARG) algorithm. We prove that ARG enjoys accelerated $O(\frac{1}{T})$ convergence rate for inclusion problems with comonotone operators (Theorem 2). Note that the lower bound $\Omega(\frac{1}{T})$ holds even for the special case of convex-concave min-max optimization (Diakonikolas, 2020; Yoon & Ryu, 2021).

Our algorithm is inspired by the *Reflected Gradient* (RG) algorithm (Chambolle & Pock, 2011; Malitsky, 2015; Cui & Shanbhag, 2016; Hsieh et al., 2019) for monotone variational inequalities. Starting at initial points $z_{-1} = z_0 \in \mathcal{Z}$, the update rule of RG with step size $\eta > 0$ is as follows: for $t = 0, 1, 2, \cdots$

$$\begin{aligned} z_{t+\frac{1}{2}} &= 2z_t - z_{t-1}, \\ z_{t+1} &= \Pi_{\mathcal{Z}}\Big[z_t - \eta F(z_{t+\frac{1}{2}})\Big]. \end{aligned} \tag{RG}$$

We propose the following Accelerated Reflected Gradient (ARG) algorithm, which is a single-call single-resolvent first-order method. Given arbitrary initial points $z_0 = z_{\frac{1}{2}} \in \mathbb{R}^n$ and step size $\eta > 0$, ARG sets $z_1 = J_{\eta A}[z_0 - \eta F(z_0)]$ and updates for $t = 1, 2, \cdots$

$$\begin{aligned} z_{t+\frac{1}{2}} &= 2z_t - z_{t-1} + \frac{1}{t+1}(z_0 - z_t) - \frac{1}{t}(z_0 - z_{t-1}), \\ z_{t+1} &= J_{\eta A}\Big[z_t - \eta F(z_{t+\frac{1}{2}}) + \frac{1}{t+1}(z_0 - z_t)\Big]. \end{aligned} \tag{ARG}$$

We use the idea from *Halpern iteration* (Halpern, 1967) to design the accelerated algorithm (ARG). This technique for deriving optimal first-order methods is also called *Anchoring* and receives intense attention recently (Diakonikolas, 2020; Yoon & Ryu, 2021; Lee & Kim, 2021a; Tran-Dinh & Luo, 2021; Tran-Dinh, 2022; Cai et al., 2022a). We defer detailed discussion on these works to Appendix A. We remark that the state of the art result from (Cai et al., 2022a) is a variant of the EG algorithm that makes two oracle calls per iteration. Thus, to the best of our knowledge, ARG is the first single-call single-resolvent algorithm with optimal convergence rate for general inclusion problems with comonotone operators.

**Theorem 2.** *Assume Assumption 1 and 2 hold for $\rho \in [-\frac{1}{60L}, 0]$, then the accelerated reflected gradient (ARG) algorithm with constant step size $\eta > 0$ satisfying Inequality (10) has the following convergence rate: for any $T \geq 1$,*

$$r_{F,A}^{tan}(z_T) \leq \frac{\sqrt{6}H}{\eta} \cdot \frac{1}{T},$$

*where $H^2 = \|z_0 - z^*\|^2 + 4\|z_1 - z_0\|^2 \leq \|z_0 - z^*\|^2 + 4r_{F,A}^{tan}(z_0)^2$.*

**Remark 1.** *Note that if Assumption 2 is satisfied with respect to some $\rho > 0$, it also satisfies Assumption 2 with $\rho = 0$, so Theorem 2 applies.*

We provide a proof sketch for Theorem 2 here and the full proof in Appendix C. Our proof is based on a potential function argument similar to the one in (Cai et al., 2022a).

**Proof Sketch.** We apply a potential function argument. We first show the potential function is approximately non-increasing and then prove that it is upper bounded by a term independent of $T$. As the potential function at step $t$ is also at least $\Omega(t^2) \cdot r^{tan}(z_t)^2$, we conclude that ARG has an $O(\frac{1}{T})$ convergence rate.

## 5 LAST-ITERATE CONVERGENCE RATE OF REFLECTED GRADIENT

In this section, we show that the *Reflected Gradient* (RG) algorithm (Chambolle & Pock, 2011; Malitsky, 2015; Cui & Shanbhag, 2016; Hsieh et al., 2019) has a last-iterate convergence rate of $O(\frac{1}{\sqrt{T}})$ with respect to tangent residual and gap function (see Definition 4) for solving monotone variational inequalities (Theorem 3).

**Theorem 3.** *For a variational inequality problem (VI) associated with a closed convex set $\mathcal{Z}$ and a monotone and L-Lipschitz operator $F$ with a solution $z^*$, the (RG) algorithm with constant step size $\eta \in (0, \frac{1}{(1+\sqrt{2})L})$ has the following last-iterate convergence rate: for any $T \geq 1$,*

$$r^{tan}_{F,\mathcal{Z}}(z_T) \leq \frac{\lambda HL}{\sqrt{T}}, \quad \mathrm{GAP}_{\mathcal{Z},F,D}(z_T) \leq \frac{\lambda DHL}{\sqrt{T}}$$

*where $H^2 = 4\|z_0 - z^*\|^2 + \frac{13}{L^2}\|F(z_0)\|^2$ and $\lambda = \sqrt{\frac{6(1+3\eta^2L^2)}{\eta^2L^2(1-(1+\sqrt{2})\eta L)}}$.*

We remark that the convergence rate of RG is slower than ARG and other optimal first-order algorithms even in the monotone setting. Nevertheless, understanding its last-iterate convergence rate is still interesting: (1) RG is simple and largely used in practice; (2) Last-iterate convergence rates of simple classic algorithms such as EG and RG are mentioned as open problems in (Hsieh et al., 2019). The question is recently resolved for EG (Gorbunov et al., 2022a; Cai et al., 2022b) but remains open for RG; (3) Compared to EG, RG requires only a single call to $F$ and a single projection in each iteration.

We provide a proof sketch for Theorem 3 here and the full proof in Appendix D.

**Proof Sketch.** Our analysis is based on a potential function argument and can be summarized in the following three steps. (1) We construct a potential function and show that it is non-increasing between two consecutive iterates; (2) We prove that the (RG) algorithm has a best-iterate convergence rate, i.e., for any $T \geq 1$, there exists one iterate $t^* \in [T]$ such that our potential function at iterate $t^*$ is small; (3) We combine the above steps to show that the the last iterate has the same convergence guarantee as the best iterate and derive the $O(\frac{1}{\sqrt{T}})$ last-iterate convergence rate.

## 6 CONCLUSION

This paper introduces single-call single-resolvent algorithms for non-monotone inclusion problems. We prove that OG has $O(\frac{1}{\sqrt{T}})$ convergence rate for problems satisfying *weak MVI* and design a new algorithm – ARG that has the optimal $O(\frac{1}{T})$ convergence rate for problems satisfying negative comonotonicity. Finally, we resolve the problem of last-iterate convergence rate of RG.

ACKNOWLEDGEMENTS

Yang Cai is supported by a Sloan Foundation Research Fellowship and the NSF Award CCF-1942583 (CAREER). We thank the anonymous reviewers for their constructive comments.

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

## CONTENTS

# A    ADDITIONAL RELATED WORKS

## A.1    CONVEX-CONCAVE AND MONOTONE SETTING

In the convex-concave setting, a weak convergence measure is the *gap* function (Definition 4). It is well-known that classic extragradient-type methods such as EG and PEG have $O(\frac{1}{T})$ average-iterate convergence rate in terms of gap function (Nemirovski, 2004; Nesterov, 2007; Mokhtari et al., 2020b; Hsieh et al., 2019) and the rate is optimal (Ouyang & Xu, 2021). But the gap function or average-iterate convergence is not meaningful in the nonconvex-nonconcave setting. For convergence in terms of the residual in the constrained setting, EG and PEG has a slower rate of $O(\frac{1}{\sqrt{T}})$ for best-iterate convergence (Korpelevich, 1976; Popov, 1980; Facchinei & Pang, 2003; Hsieh et al., 2019) and the more desirable last-iterate convergence (Cai et al., 2022b; Gorbunov et al., 2022b). We remark that the last-iterate convergence rate of the reflected gradient (RG) algorithm was unknown. The $O(\frac{1}{\sqrt{T}})$ rate is tight for $p$-SCLI algorithms (Golowich et al., 2020), a subclass of first-order methods that includes EG, PEG, and many of its variations, but faster rate is possible for other first-order methods.

**Accelerated Convergence Rate in Residual.**    Recent results with accelerated convergence rates in terms of the residual are based on Halpern iteration (Halpern, 1967) (also called *Anchoring*). The vanilla Halpern iteration has $O(\frac{1}{T})$ convergence rate for cocoercive operators (stronger than monotonicity) (Diakonikolas, 2020; Kim, 2021). Recently, a line of works contributed to provide $O(\frac{1}{T})$ convergence rate for monotone operators in the constrained setting. Diakonikolas (2020); Yoon & Ryu (2022) provide double-loop algorithms with $O(\frac{\log T}{T})$ convergence rate for monotone operators in the constrained setting. In the unconstrained setting ($A = 0$), Yoon & Ryu (2021) propose the Extra Anchored Gradient (EAG) algorithm, the first efficient algorithm with $O(\frac{1}{T})$ convergence rate for monotone operators. They also establish matching lower bound for first-order methods. Lee & Kim (2021a) generalize EAG to Fast Extragradient (FEG), which works even for negatively comonotone operators but still in the unconstrained setting. Analysis for variants of EAG and FEG in the unconstrained setting is provided in (Tran-Dinh & Luo, 2021; Tran-Dinh, 2022). Recently, Cai et al. (2022a) close the open problem by proving the projected version of EAG has $O(\frac{1}{T})$ convergence rate. They also propose the accelerated forward-backward splitting (AS) algorithm, a generalization of FEG, which has $O(\frac{1}{T})$ convergence rate for negatively comonotone operators in the constrained setting.

## A.2    NONCONVEX-NONCONCAVE SETTING

This paper study structured nonconvex-nonconcave optimization problems from the general perspective of operator theory and focus on global convergence under weak MVI and negative comonotonicity. There is a line of works focusing on *local convergence*, e.g., (Heusel et al., 2017; Mazumdar et al., 2019; Jin et al., 2020; Fiez & Ratliff, 2021). Another line of works focus on problems satisfying different structural assumptions, such as the Polyak Łojasiewicz condition (Nouiehed et al., 2019; Yang et al., 2020).

# B    ADDITIONAL PRELIMINARY

## B.1    RESOLVENT AND PROXIMAL OPERATOR

When $A = \partial g$ is the subdifferential operator of a lower semi-continuous, proper, and convex function $f$, its resolvent $(I + \lambda \partial g)^{-1}$ is also known as the *proximal operator* of $g$ denoted as $\mathbf{prox}_{\lambda g}$. The resolvent $(I + \lambda \partial g)^{-1}$ is efficiently computable for the following popular choices of function $g$: $\ell_1$-norm $|| \cdot ||_1$, $\ell_2$-norm $|| \cdot ||_2$, maxtrix norms, the log-barrier $-\sum_{i=1}^{n} \log(x_i)$, and more generally any quadratic or smooth functions. Moreover, many of them have closed-form expressions. For example, the proximal operator of the $\ell_1$-norm $g =$

$||\cdot||_1$ is the element-wise *soft-thresholding* operator $(\mathbf{prox}_{\lambda g}(v))_i = (v_i - \lambda)_+ - (-v_i - \lambda)_+$.
We refer readers to (Parikh & Boyd, 2014, Chapter 6, 7) for a comprehensive review on
proximal operators and their efficient computation.

### B.2 GAP FUNCTION

A standard suboptimality measure for the variationaly inequalitt (VI) problem is the *gap
function* defined as $\mathrm{GAP}_{\mathcal{Z},F}(z) := \max_{z' \in \mathcal{Z}} \langle F(z), z - z' \rangle$. Note that when the feasible set $\mathcal{Z}$
is unbounded, approximating the gap function is impossible: consider the simple uncon-
strained saddle point problem $\min_{x \in \mathbb{R}} \max_{y \in \mathbb{R}} xy$, which has a unique saddle point $(0,0)$
but any other point has an infinitely large gap. A refined notion is the following restricted
gap function (Nesterov, 2007), which is meaningful for unbounded $\mathcal{Z}$.

**Definition 4** (Restricted Gap Function). *Given a closed convex set $\mathcal{Z}$, a single-valued operator
F, and a radius D, the restricted gap function at point $z \in \mathcal{Z}$ is*

$$\mathrm{GAP}_{\mathcal{Z},F,D} := \max_{z' \in \mathcal{Z} \cap \mathcal{B}(z,D)} \langle F(z), z - z' \rangle$$

*where $\mathcal{B}(z, D)$ is a Euclidean ball centered at z with radius D.*

In the rest of the paper, we call $\mathrm{GAP}_{\mathcal{Z},F,D}$ the gap function (or gap) for convenience. The
following Lemma relates $\|F(z) + c\|$ where $c \in N_{\mathcal{Z}}(z)$, and the gap function.

**Lemma 1.** *Let $\mathcal{Z}$ be a closed convex set $\mathcal{Z}$ and F be a monotone and L-Lipschitz operator. For any
$z \in \mathcal{Z}$ and $c \in N_{\mathcal{Z}}(z)$, we have*

$$\mathrm{GAP}_{\mathcal{Z},F,D}(z) := \max_{z' \in \mathcal{Z} \cap \mathcal{B}(z,D)} \langle F(z), z - z' \rangle \leq D \cdot \|F(z) + c\|.$$

*Proof.* The proof is straightforward. Since $c \in N_{\mathcal{Z}}(z)$, we have $\langle c, z - z' \rangle \geq 0$ for any
$z' \in \mathcal{Z}$. Therefor,

$$\max_{z' \in \mathcal{Z} \cap \mathcal{B}(z,D)} \langle F(z), z - z' \rangle \leq \max_{z' \in \mathcal{Z} \cap \mathcal{B}(z,D)} \langle F(z) + c, z - z' \rangle$$

$$\leq \max_{z' \in \mathcal{Z} \cap \mathcal{B}(z,D)} \|z - z'\| \cdot \|F(z) + c\|$$

(Cauchy-Schwarz inequality)

$$\leq D \cdot \|F(z) + c\|.$$

$\square$

### B.3 CLASSICAL ALGORITHMS FOR VARIATIONALY INEQUALITIES

**The Extragradient Algorithm (Korpelevich, 1976).** Starting at initial point $z_0 \in \mathcal{Z}$, the
update rule of EG is: for $t = 0, 1, 2, \cdots$

$$z_{t+\frac{1}{2}} = \Pi_{\mathcal{Z}}[z_t - \eta F(z_t)],$$
$$z_{t+1} = \Pi_{\mathcal{Z}}\left[z_t - \eta F(z_{t+\frac{1}{2}})\right]. \tag{EG}$$

At each step $t \geq 0$, the EG algorithm makes an oracle call of $F(z_t)$ to produce an interme-
diate point $z_{t+\frac{1}{2}}$ (a gradient descent step if $F = \partial f$ is the gradient of some function $f$), then
the algorithm makes another oracle call $F(z_{t+\frac{1}{2}})$ and updates $z_t$ to $z_{t+1}$. In each step, EG
needs two oracle calls to F and two projections $\Pi_{\mathcal{Z}}$.

**The Past Extragradient Algorithm (Popov, 1980)** Starting at initial point $z_0 = z_{-\frac{1}{2}} \in \mathcal{Z}$,
the update rule of PEG with step size $\eta > 0$ is: for $t = 0, 1, 2, \cdots$

$$z_{t+\frac{1}{2}} = \Pi_{\mathcal{Z}}\left[z_t - \eta F(z_{t-\frac{1}{2}})\right],$$
$$z_{t+1} = \Pi_{\mathcal{Z}}\left[z_t - \eta F(z_{t+\frac{1}{2}})\right]. \tag{PEG}$$

Note that PEG is also known as the *Optimistic Gradient Descent/Ascent (OGDA)* algorithm in the literature. The update rule of PEG is similar to (EG) but only requires a single call to $F$ in each iteration. Both of EG and PEG perform two projections in every iteration.

### B.4 TANGENT RESIDUAL UPPER BOUNDS OTHER NOTIONS OF RESIDUAL

**Proposition 2.** *Let $A$ be a maximally monotone operator and $F$ be an single-valued operator. Then for any $z \in \mathbb{R}^n$ and $\alpha > 0$,*

$$r_{F,A}^{tan}(z) \geq r_{F,A}^{nat}(z) := \|z - J_A(z - F(z))\|$$

$$r_{F,A}^{tan}(z) \geq r_{F,A,\alpha}^{fb}(z) := \frac{1}{\alpha}\|z - J_{\alpha A}[z - \alpha F(z)]\|.$$

*Proof.* For any $c \in A(z)$, we have

$$\begin{aligned}
r_{F,A}^{nat}(z) &= \|z - J_A(z - F(z))\| \\
&= \|J_A(z + c) - J_A(z - F(z))\| \\
&\leq \|F(z) + c\| \qquad\qquad (J_A \text{ is non-expansive})
\end{aligned}$$

and

$$\begin{aligned}
r_{F,A,\alpha}^{fb}(z) &= \frac{1}{\alpha}\|z - J_{\alpha A}(z - \alpha F(z))\| \\
&= \frac{1}{\alpha}\|J_{\alpha A}(z + \alpha c) - J_{\alpha A}(z - \alpha F(z))\| \\
&\leq \|F(z) + c\|. \qquad\qquad (J_A \text{ is non-expansive})
\end{aligned}$$

Thus both $r_{F,A}^{tan}(z)$ and $r_{F,A,\alpha}^{fb}(z)$ are smaller than $r_{F,A}^{tan}(z) = \min_{c \in A(z)} \|F(z) + c\|$. $\qquad\square$

## C MISSING PROOFS IN SECTION 4

To prove Theorem 2, we apply a potential function argument. We first show the potential function is approximately non-increasing and then prove that it is upper bounded by a term independent of $T$. As the potential function at step $t$ is also at least $\Omega(t^2) \cdot r^{tan}(z_t)^2$, we conclude that ARG has an $O(\frac{1}{T})$ convergence rate .

### C.1 POTENTIAL FUNCTION

Recall the update rule of ARG: $z_0 = z_{\frac{1}{2}} \in \mathbb{R}^n$ are initial points and $z_1 = J_{\eta A}[z_0 - \eta F(z_0)]$; for $t \geq 1$,

$$\begin{aligned}
z_{t+\frac{1}{2}} &= 2z_t - z_{t-1} + \frac{1}{t+1}(z_0 - z_t) - \frac{1}{t}(z_0 - z_{t-1}), \\
z_{t+1} &= J_{\eta A}\left[z_t - \eta F(z_{t+\frac{1}{2}}) + \frac{1}{t+1}(z_0 - z_t)\right].
\end{aligned} \qquad\text{(ARG)}$$

Recall that when $A$ is the normal cone of a closed convex set $\mathcal{Z}$, the resolvent $J_A$ is equivalent to Euclidean projection to set $\mathcal{Z}$. Hence, if we apply the ARG algorithm to solve monotone VI problems, the algorithm uses a single call to operator $F$ and a single projection to $\mathcal{Z}$ per iteration. Here we allow $A$ to be an arbitrary maximally monotone operator, and the ARG algorithm becomes a single-call single-resolvent algorithm in this more general setting.

Next, we specify the potential function. Define

$$c_{t+1} := \frac{z_t - \eta F(z_{t+\frac{1}{2}}) + \frac{1}{t+1}(z_0 - z_t) - z_{t+1}}{\eta}, \quad \forall t \geq 0. \qquad (8)$$

By update rule we have $c_t \in A(z_t)$ for all $t \geq 1$. The potential function at iterate $t \geq 1$ is defined as

$$V_t := \frac{t(t+1)}{2} \|\eta F(z_t) + \eta c_t\|^2 + \frac{t(t+1)}{2} \left\|\eta F(z_t) - \eta F(z_{t-\frac{1}{2}})\right\|^2 + t\langle \eta F(z_t) + \eta c_t, z_t - z_0 \rangle. \tag{9}$$

## C.2 APPROXIMATELY NON-INCREASING POTENTIAL

**Fact 2.** *For any $L > 0$ and $\rho \geq -\frac{1}{60L}$. There exists $\eta > 0$ such that*

$$\frac{1}{2} - (12 - \frac{4\rho}{\eta})\eta^2 L^2 + \frac{2\rho}{\eta} \geq 0. \tag{10}$$

*Moreover, every $\eta > 0$ satisfies (10) also satisfies $\frac{\rho}{\eta} \geq -\frac{1}{4}$.*

*Proof.* Rewriting (10), we get

$$\rho > \frac{\eta L(24\eta^2 L^2 - 1)}{4 + 8\eta^2 L^2} \cdot \frac{1}{L}.$$

Let $x = \eta L$ and $f(x) = \frac{x(24x^2 - 1)}{4 + 8x^2}$. Since $f(\frac{1}{12}) = -\frac{5}{292} < -\frac{1}{60}$. We know $\eta = \frac{1}{12L}$ satisfies (10).

Moreover, rewritng (10) and using $\eta L > 0$, we get

$$\frac{\rho}{\eta} \geq -\frac{1 - 72\eta^2 L^2}{4 + 8\eta^2 L^2} \geq -\frac{1}{4}.$$

$\square$

We show in the following lemma that $V_t$ is approximately non-increasing.

**Lemma 2.** *In the same setup as Theorem 2, for any $t \geq 1$, we have*

$$V_{t+1} \leq V_t + \frac{1}{8} \cdot \|\eta F(z_{t+1}) + \eta c_{t+1}\|^2.$$

*Proof.* The plan is to show that $V_t - V_{t+1}$ plus a few non-positive terms is still $\geq -\frac{1}{8} \cdot \|\eta F(z_{t+1}) + \eta c_{t+1}\|^2$, which certifies the claim.

**Two Positive Terms.** Since $F + A$ is $\rho$-comonotone, we have

$$\langle \eta F(z_{t+1}) + \eta c_{t+1} - \eta F(z_t) - \eta c_t, z_{t+1} - z_t \rangle - \frac{\rho}{\eta} \|\eta F(z_{t+1}) + \eta c_{t+1} - \eta F(z_t) - \eta c_t\|^2 \geq 0. \tag{11}$$

Since $F$ is $L$-Lipschitz, we have

$$\eta^2 L^2 \cdot \left\|z_{t+1} - z_{t+\frac{1}{2}}\right\|^2 - \left\|\eta F(z_{t+1}) - \eta F(z_{t+\frac{1}{2}})\right\|^2 \geq 0.$$

Denote $p = \frac{1}{24}$. Multiplying the above inequality with $1 - \frac{\rho}{3\eta} > 0$ and rearranging terms, we get

$$p \cdot \left\|z_{t+1} - z_{t+\frac{1}{2}}\right\|^2 - \left\|\eta F(z_{t+1}) - \eta F(z_{t+\frac{1}{2}})\right\|^2$$
$$+ \left((1 - \frac{\rho}{3\eta})\eta^2 L^2 - p\right) \cdot \left\|z_{t+1} - z_{t+\frac{1}{2}}\right\|^2 + \frac{\rho}{3\eta}\left\|\eta F(z_{t+1}) - \eta F(z_{t+\frac{1}{2}})\right\|^2 \geq 0. \tag{12}$$

**Sum-of-Squares Identity.** We show an equivalent formulation $z_{t+\frac{1}{2}}$ and $z_{t+1}$ using definitions of $\eta c_t = z_{t-1} - z_t - \eta F(z_{t-\frac{1}{2}}) + \frac{1}{t}(z_0 - z_{t-1})$ and $\eta c_{t+1} = z_t - \eta F(z_{t+\frac{1}{2}}) + \frac{1}{t+1}(z_0 - z_t) - z_{t+1}$:

$$z_{t+\frac{1}{2}} = 2z_t - z_{t-1} + \frac{1}{t+1}(z_0 - z_t) - \frac{1}{t}(z_0 - z_{t-1})$$

$$= z_t + (z_t - z_{t-1}) + \frac{1}{t+1}(z_0 - z_t) - \frac{1}{t}(z_0 - z_{t-1})$$

$$= z_t - \eta F(z_{t-\frac{1}{2}}) - \eta c_t + \frac{1}{t+1}(z_0 - z_t),$$

$$z_{t+1} = z_t - \eta F(z_{t+\frac{1}{2}}) - \eta c_{t+1} + \frac{1}{t+1}(z_0 - z_t).$$

We also have

$$z_{t+1} - z_{t+\frac{1}{2}} = \eta F(z_{t-\frac{1}{2}}) + \eta c_t - \eta F(z_{t+\frac{1}{2}}) - \eta c_{t+1}. \tag{13}$$

Next, we simplify

$$V_t - V_{t+1} - t(t+1) \times \text{LHS of Inequality (11)} - \frac{t(t+1)}{4p} \times \text{LHS of Inequality (12)}$$

using the second identity in Proposition 3: replace $x_0$ with $z_0$; for $k \in [4]$, replace $x_k$ with $z_{t-1+\frac{k}{2}}$ and replace $y_k$ with $\eta F(z_{t-1+\frac{k}{2}})$; replace $u_2$ with $\eta c_t$; replace $u_4$ with $\eta c_{t+1}$; replace $k$ with $t$; replace $p$ with $q$. Note that $x_3 = x_2 - y_1 - u_2 + \frac{1}{k+1}(x_0 - x_2)$ and $x_4 = x_2 - y_3 - u_4 + + \frac{1}{k+1}(x_0 - x_2)$ hold due to the above equivalent formations of $z_{t+\frac{1}{2}}$ and $z_{t+1}$. Expression (17) and (18) appear on both sides of the following equation.

$$V_t - V_{t+1} - t(t+1) \times \text{LHS of Inequality (11)} - \frac{t(t+1)}{4p} \times \text{LHS of Inequality (12)}$$

$$= \frac{t(t+1)}{4}\left\|\eta c_{t+1} - \eta c_t + \eta F(z_{t-\frac{1}{2}}) - 2\eta F(z_t) + \eta F(z_{t+\frac{1}{2}})\right\|^2 \tag{14}$$

$$+ \left(\frac{(1-4p)t - 4p}{4p}(t+1)\right) \cdot \left\|\eta F(z_{t+\frac{1}{2}}) - \eta F(z_{t+1})\right\|^2 \tag{15}$$

$$+ (t+1) \cdot \left\langle \eta F(z_{t+\frac{1}{2}}) - \eta F(z_{t+1}), \eta F(z_{t+1}) + \eta c_{t+1} \right\rangle \tag{16}$$

$$+ t(t+1)\frac{\rho}{\eta} \cdot \|\eta F(z_{t+1}) + \eta c_{t+1} - \eta F(z_t) - \eta c_t\|^2 \tag{17}$$

$$- \frac{t(t+1)}{4p} \cdot \left(\left((1 - \frac{\rho}{3\eta})\eta^2 L^2 - p\right) \cdot \left\|z_{t+1} - z_{t+\frac{1}{2}}\right\|^2 + \frac{\rho}{3\eta}\left\|\eta F(z_{t+1}) - \eta F(z_{t+\frac{1}{2}})\right\|^2\right). \tag{18}$$

Since $\|a\|^2 + \langle a, b \rangle = \|a + \frac{b}{2}\|^2 - \frac{\|b\|^2}{4}$, we have

Expression (15) + Expression (16)

$$= \left\|\sqrt{\frac{(1-4p)t - 4p}{4p}(t+1)} \cdot \left(\eta F(z_{t+\frac{1}{2}}) - \eta F(z_{t+1})\right) + \sqrt{\frac{p(t+1)}{(1-4p)t - 4p}} \cdot (\eta F(z_{t+1}) + \eta c_{t+1})\right\|^2$$

$$- \frac{p(t+1)}{(1-4p)t - 4p} \cdot \|\eta F(z_{t+1}) + \eta c_{t+1}\|^2$$

$$\geq -\frac{p(t+1)}{(1-8p)t} \cdot \|\eta F(z_{t+1}) + \eta c_{t+1}\|^2 \qquad (t \geq 1)$$

$$\geq -\frac{2p}{1-8p} \cdot \|\eta F(z_{t+1}) + \eta c_{t+1}\|^2 \qquad (\tfrac{t+1}{t} \leq 2)$$

$$= -\frac{1}{8}\|\eta F(z_{t+1}) + \eta c_{t+1}\|^2. \qquad (p = \tfrac{1}{24})$$

Now it remains to show that the sum of Expression (14), (17), and (18) is non-negative. Multiplying $\frac{4}{t(t+1)}$ and replacing $p = \frac{1}{24}$, we get

$$\frac{4}{t(t+1)} \cdot (\text{Expression (14)} + \text{Expression (17)} + \text{Expression (18)})$$

$$= \left\| \eta c_{t+1} - \eta c_t + \eta F(z_{t-\frac{1}{2}}) - 2\eta F(z_t) + \eta F(z_{t+\frac{1}{2}}) \right\|^2 + \left( 1 - (24 - \frac{8\rho}{\eta})\eta^2 L^2 \right) \cdot \left\| z_{t+1} - z_{t+\frac{1}{2}} \right\|^2$$

$$+ \frac{4\rho}{\eta} \cdot \left\| \eta F(z_{t+1}) + \eta c_{t+1} - \eta F(z_t) - \eta c_t \right\|^2 - \frac{8\rho}{\eta} \left\| \eta F(z_{t+1}) - \eta F(z_{t+\frac{1}{2}}) \right\|^2.$$

Denote

$$\begin{aligned}
B_1 &= \eta c_{t+1} - \eta c_t + \eta F(z_{t-\frac{1}{2}}) - 2\eta F(z_t) + \eta F(z_{t+\frac{1}{2}}) \\
B_2 &= z_{t+1} - z_{t+\frac{1}{2}} = \eta F(z_{t-\frac{1}{2}}) + \eta c_t - \eta F(z_{t+\frac{1}{2}}) - \eta c_{t+1} \qquad \text{(By (13))} \\
B_3 &= \eta F(z_{t+1}) + \eta c_{t+1} - \eta F(z_t) - \eta c_t \\
B_4 &= \eta F(z_{t+1}) - \eta F(z_{t+\frac{1}{2}}).
\end{aligned}$$

It is not hard to check that $B_1 - B_2 = 2(B_3 - B_4)$:

$$B_1 - B_2 = 2\eta c_{t+1} - 2\eta c_t - 2\eta F(z_t) + 2\eta F(z_{t+\frac{1}{2}}) = 2(B_3 - B_4).$$

Note that $\rho$ is non-positive and we have

$$\frac{4}{t(t+1)} \cdot (\text{Expression (14)} + \text{Expression (17)} + \text{Expression (18)})$$

$$= \|B_1\|^2 + \left( 1 - (24 - \frac{8\rho}{\eta})\eta^2 L^2 \right) \cdot \|B_2\|^2 + \frac{\rho}{\eta} \cdot \|2B_3\|^2 - \frac{2\rho}{\eta}\|2B_4\|^2$$

$$\geq \left( \frac{1}{2} - (12 - \frac{4\rho}{\eta})\eta^2 L^2 \right) \cdot \|B_1 - B_2\|^2 + \frac{\rho}{\eta} \cdot \|2B_3\|^2 - \frac{2\rho}{\eta}\|2B_4\|^2$$

$$\qquad\qquad (\|a\|^2 + \|b\|^2 \geq \tfrac{1}{2}\|a - b\|^2 \text{ and } (24 - \tfrac{8\rho}{\eta})\eta^2 L^2 \geq 0)$$

$$\geq \left( \frac{1}{2} - (12 - \frac{4\rho}{\eta})\eta^2 L^2 \right) \cdot \|B_1 - B_2\|^2 + \frac{2\rho}{\eta} \cdot \|2B_3 - 2B_4\|^2$$

$$\qquad\qquad (-\|a\|^2 + 2\|b\|^2 \geq -2\|a - b\|^2 \text{ and } -\tfrac{\rho}{\eta} \geq 0)$$

$$= \left( \frac{1}{2} - (12 - \frac{4\rho}{\eta})\eta^2 L^2 + \frac{2\rho}{\eta} \right) \cdot \|B_1 - B_2\|^2 \qquad (B_1 - B_2 = 2(B_3 - B_4))$$

$$\geq 0. \qquad\qquad\qquad\qquad\qquad\qquad\qquad\qquad\qquad\qquad\qquad\qquad (\text{Inequality (10)})$$

The last inequality holds by the choice of $\eta$ as shown in Fact 2. $\qquad\square$

## C.3  BOUDING POTENTIAL AT ITERATION 1

**Lemma 3.** *Let $F$ be a $L$-Lipschitz operator and $A$ be a maximally monotone operator. For any $z_0 = z_{\frac{1}{2}} \in \mathbb{R}^n$, $\eta \in (0, \frac{1}{2L})$, and $z_1 = J_{\eta A}[z_0 - \eta F(z_0)]$, we have the following*

1. $\|z_1 - z_0\| \leq \eta \cdot r_{F,A}^{tan}(z_0)$.

2. $\|\eta F(z_1) + \eta c_1\| \leq (1 + \eta L)\|z_1 - z_0\|$.

3. $V_1 \leq 4\|z_1 - z_0\|^2$ *where $V_1$ is defined in (9).*

*Proof.* For any $c \in A(z_0)$, due to non-expansiveness of $J_{\eta A}$, we have

$$\|z_1 - z_0\| = \left\| J_{\eta A}[z_0 - \eta F(z_0)] - J_{\eta A}[z_0 + \eta c] \right\| \leq \eta \|F(z_0) + c\|.$$

Thus $\|z_1 - z_0\| \leq \eta \cdot r_{F,A}^{tan}(z_0)$.

By definition of $V_1$ in (9), we have

$$V_1 = \|\eta F(z_1) + \eta c_1\|^2 + \|\eta F(z_1) - \eta F(z_0)\|^2 + \langle \eta F(z_1) + \eta c_1, z_1 - z_0 \rangle.$$

We bound $\|\eta F(z_1) + \eta c_1\|$ first. Note that by definition, we have $\eta c_1 = z_0 - \eta F(z_0) - z_1$. Thus we have

$$\begin{aligned}
\|\eta F(z_1) + \eta c_1\| &= \|z_0 - z_1 + \eta F(z_1) - \eta F(z_0)\| \\
&\leq \|z_0 - z_1\| + \|\eta F(z_1) - \eta F(z_0)\| &\text{(triangle inequality)} \\
&\leq (1 + \eta L)\|z_1 - z_0\|. &\text{($F$ is $L$-Lipschitz)}
\end{aligned}$$

Then we can apply the bound on $\|\eta F(z_1) + \eta c_1\|$ to bound $V_1$ as follows:

$$\begin{aligned}
V_1 &= \|\eta F(z_1) + \eta c_1\|^2 + \|\eta F(z_1) - \eta F(z_0)\|^2 + \langle \eta F(z_1) + \eta c_1, z_1 - z_0 \rangle \\
&\leq \|\eta F(z_1) + \eta c_1\|^2 + \eta^2 L^2 \|z_1 - z_0\|^2 + \|\eta F(z_1) + \eta c_1\|\|z_1 - z_0\| \\
&\leq (1 + \eta L)^2 \|z_1 - z_0\|^2 + \eta^2 L^2 \|z_1 - z_0\|^2 + (1 + \eta L)\|z_1 - z_0\|^2 \\
&= (2 + 3\eta L + 2\eta^2 L^2)\|z_1 - z_0\|^2 \\
&\leq 4\|z_1 - z_0\|^2.
\end{aligned}$$

where we use $L$-Lipschitzness of $F$ and Cauchy-Schwarz inequality in the first inequality; we use $\|\eta F(z_1) + \eta c_1\| \leq (1 + \eta L)\|z_1 - z_0\|$ in the second inequality; we use $\eta L \leq \frac{1}{2}$ in the last inequality. $\qquad\square$

### C.4 Proof of Theorem 2

We first show that the potential function $V_t = \Omega(t^2 \cdot r^{tan}(z_t)^2)$.

**Lemma 4.** *In the same setup as Theorem 2, for any $t \geq 1$, we have*

$$\frac{t(t + \frac{1}{2})}{4} \|\eta F(z_t) + \eta c_t\|^2 \leq V_t + \|z^* - z_0\|^2.$$

*Proof.* Since $0 \in F(z^*) + A(z^*)$, by $\rho$-comonotonicity of $F + A$ and Fact 2, we have

$$\langle \eta F(z_t) + \eta c_t, z_t - z^* \rangle \geq \frac{\rho}{\eta}\|\eta F(z_t) + \eta c_t\|^2 \geq -\frac{1}{4}\|\eta F(z_t) + \eta c_t\|^2. \qquad (19)$$

By definition of $V_t$ in (9), for any $t \geq 1$, we have

$$\begin{aligned}
V_t &= \frac{t(t+1)}{2}\|\eta F(z_t) + \eta c_t\|^2 + \frac{t(t+1)}{2}\left\|\eta F(z_t) - \eta F(z_{t-\frac{1}{2}})\right\|^2 + t\langle \eta F(z_t) + \eta c_t, z_t - z_0 \rangle \\
&\geq \frac{t(t+1)}{2}\|\eta F(z_t) + \eta c_t\|^2 + t\langle \eta F(z_t) + \eta c_t, z_t - z^* \rangle + t\langle \eta F(z_t) + \eta c_t, z^* - z_0 \rangle \\
&\geq \frac{t(t+1)}{2}\|\eta F(z_t) + \eta c_t\|^2 - \frac{1}{4}\|\eta F(z_t) + \eta c_t\|^2 + t\langle \eta F(z_t) + \eta c_t, z^* - z_0 \rangle \\
&\qquad\qquad\qquad\qquad\qquad\qquad\qquad\qquad\qquad\qquad\qquad \text{(By Inequality (19))} \\
&\geq \frac{t(t+\frac{1}{2})}{2}\|\eta F(z_t) + \eta c_t\|^2 - \frac{t(t+\frac{1}{2})}{4}\|\eta F(z_t) + \eta c_t\|^2 - \frac{t}{t+\frac{1}{2}}\|z^* - z_0\|^2 \\
&\geq \frac{t(t+\frac{1}{2})}{4}\|\eta F(z_t) + \eta c_t\|^2 - \|z^* - z_0\|^2 &(\tfrac{t}{t+\frac{1}{2}} < 1)
\end{aligned}$$

where in the second last inequality we we apply $\langle a, b \rangle \geq -\frac{\alpha}{4}\|a\|^2 - \frac{1}{\alpha}\|b\|^2$ with $a = \sqrt{t}(\eta F(z_t) + \eta c_t)$, $b = \sqrt{t}(z^* - z_0)$, and $\alpha = t + \frac{1}{2}$. $\qquad\square$

*Proof of Theorem 2.* It is equivalent to prove that for every $T \geq 1$, we have

$$\|\eta F(z_T) + \eta c_T\|^2 \leq \frac{6H^2}{T^2}.$$

From Lemma 3, we have

$$\|\eta F(z_1) + \eta c_1\|^2 \leq (1 + \eta L)^2 \|z_1 - z_0\|^2 \leq H^2.$$

So the theorem holds for $T = 1$.

For any $T \geq 2$, by Lemma 4 we have

$$\frac{T(T + \frac{1}{2})}{4} \|\eta F(z_T) + \eta c_T\|^2 \leq V_T + \|z_0 - z^*\|^2$$

$$\leq V_1 + \|z_0 - z^*\|^2 + \frac{1}{8} \sum_{t=2}^{T} \|\eta F(z_t) + \eta c_t\|^2$$

$$= H^2 + \frac{1}{8} \sum_{t=2}^{T} \|\eta F(z_t) + \eta c_t\|^2.$$

By subtracting $\frac{1}{8} \|\eta F(z_T) + \eta c_T\|^2$ from both sides of the above inequality, we get

$$\frac{T^2}{4} \|\eta F(z_T) + \eta c_T\|^2 \leq H^2 + \frac{1}{8} \sum_{t=2}^{T-1} \|\eta F(z_t) + \eta c_t\|^2$$

which is in the form of Proposition 4 with $C_1 = H^2$ and $p = \frac{1}{9}$. Thus we have for any $T \geq 2$

$$\|\eta F(z_T) + \eta c_T\|^2 \leq \frac{6H^2}{T^2}.$$

$\square$

## D  MISSING PROOFS IN SECTION 5

To prove Theorem 3, our analysis is based on a potential function argument and can be summarized in the following three steps. (1) We construct a potential function and show that it is non-increasing between two consecutive iterates; (2) We prove that the RG algorithm has a best-iterate convergence rate, i.e., for any $T \geq 1$, there exists one iterate $t^* \in [T]$ such that our potential function at iterate $t^*$ is small; (3) We combine the above steps to show that the the last iterate has the same convergence guarantee as the best iterate and derive the $O(\frac{1}{\sqrt{T}})$ last-iterate convergence rate.

### D.1  NON-INCREASING POTENTIAL

**Potential Function.**  We denote

$$c_{t+1} := \frac{z_t - \eta F(z_{t+\frac{1}{2}}) - z_{t+1}}{\eta}, \ \forall t \geq 0 \tag{20}$$

Note that according to the update rule of RG, $z_{t+1} = \Pi_{\mathcal{Z}}[z_t - \eta F(z_{t+\frac{1}{2}})]$, so $c_{t+1} \in N_{\mathcal{Z}}(z_{t+1})$.

The potential function we adopt is $P_t$ defined as

$$P_t := \|F(z_t) + c_t\|^2 + \left\|F(z_t) - F(z_{t-\frac{1}{2}})\right\|^2, \ \forall t \geq 1. \tag{21}$$

**Lemma 5.** *In the same setup of Theorem 3, $P_t \geq P_{t+1}$ for any $t \geq 1$.*

*Proof.* The plan is to show that $P_t - P_{t+1}$ plus a few non-positive terms is non-negative, which certifies that $P_t - P_{t+1} \geq 0$.

**Three Non-Positive Terms.** Since $F$ is monotone, we have

$$(-2) \cdot \langle \eta F(z_{t+1}) - \eta F(z_t), z_{t+1} - z_t \rangle \leq 0. \tag{22}$$

Since $F$ is $L$-Lipschitz and $0 < \eta < \frac{1}{(1+\sqrt{2})L} < \frac{1}{2L}$, we have

$$(-2) \cdot \left( \frac{1}{4} \cdot \left\| z_{t+1} - z_{t+\frac{1}{2}} \right\|^2 - \left\| \eta F(z_{t+1}) - \eta F(z_{t+\frac{1}{2}}) \right\|^2 \right) \leq 0. \tag{23}$$

By definition, we have $c_{t+1} \in N_{\mathcal{Z}}(z_{t+1})$ and $c_t \in N_{\mathcal{Z}}(z_t)$. Since the normal cone operator $N_{\mathcal{Z}}$ is maximally monotone, we have

$$(-2) \cdot \langle \eta c_{t+1} - \eta c_t, z_{t+1} - z_t \rangle \leq 0. \tag{24}$$

**Sum-of-Squares Identity.** We use the following equivalent formations of $z_{t+\frac{1}{2}}$ and $z_{t+1}$.

$$z_{t+\frac{1}{2}} = 2z_t - z_{t-1} = z_t - (z_{t-1} - z_t) = z_t - \eta F(z_{t-\frac{1}{2}}) - \eta c_t,$$

$$z_{t+1} = \Pi_{\mathcal{Z}} \left[ z_t - \eta F(z_{t+\frac{1}{2}}) \right] = z_t - \eta F(z_{t+\frac{1}{2}}) - \eta c_{t+1}.$$

The following identity holds according to Proposition 3. To see this, we replace $x_k$ with $z_{t-1+\frac{k}{2}}$; replace $y_k$ with $\eta F(z_{t-1+\frac{k}{2}})$; replace $u_2$ with $\eta c_t$; replace $u_4$ with $\eta c_{t+1}$; also note that $x_3 = x_2 - y_1 - u_2$ and $x_4 = x_2 - y_3 - u_4$ hold due to the above equivalent formations of $z_{t+\frac{1}{2}}$ and $z_{t+1}$.

$$\eta^2 \cdot (P_t - P_{t+1}) + \text{LHS of Inequality}(22) + \text{LHS of Inequality}(23) + \text{LHS of Inequality}(24)$$

$$= \left\| \frac{z_{t+\frac{1}{2}} - z_{t+1}}{2} + \eta F(z_{t-\frac{1}{2}}) - \eta F(z_t) \right\|^2 + \left\| \frac{z_{t+\frac{1}{2}} + z_{t+1}}{2} - z_t + \eta F(z_t) + \eta c_t \right\|^2.$$

The right-hand side of the above equality is clearly $\geq 0$, thus we conclude $P_t - P_{t+1} \geq 0$. $\square$

## D.2 BEST-ITERATE CONVERGENCE

In this section, we show that for any $T \geq 1$, there exists some iterate $t^*$ such that $P_{t^*} = O(\frac{1}{T})$, which is implied by $\sum_{t=1}^{T} P_t = O(1)$. To prove this, we first show $\sum_{t=1}^{T} \left\| z_{t+\frac{1}{2}} - z_t \right\|^2 = \sum_{t=1}^{T} \left\| z_t - z_{t-1} \right\|^2 = O(1)$ and then relate $\sum_{t=1}^{T} P_t$ to these two quantities.

**Lemma 6.** *In the same setup of Theorem 3, for any $T \geq 1$, we have*

$$\sum_{t=1}^{T} \left\| z_{t+\frac{1}{2}} - z_t \right\|^2 = \sum_{t=1}^{T} \left\| z_t - z_{t-1} \right\|^2 \leq \frac{H^2}{1 - (1+\sqrt{2})\eta L}.$$

*Proof.* First note that by the update rule of RG, we have $z_{t+\frac{1}{2}} = 2z_t - z_{t-1}$ thus $z_{t+\frac{1}{2}} - z_t = z_t - z_{t-1}$. Therefore, it suffices to only prove the inequality for $\sum_{t=1}^{T} \left\| z_{t+\frac{1}{2}} - z_t \right\|^2$.

From the proof of (Hsieh et al., 2019, Lemma 2), for any $t \geq 1$ and $p \in \mathcal{Z}$, we have

$$\left(1 - (1+\sqrt{2})\eta L\right) \cdot \left\| z_{t+\frac{1}{2}} - z_t \right\|^2 \leq \|z_t - p\|^2 - \|z_{t+1} - p\|^2 - 2\eta \left\langle F(z_{t+\frac{1}{2}}), z_{t+\frac{1}{2}} - p \right\rangle$$

$$+ \eta L \left( \left\| z_t - z_{t-\frac{1}{2}} \right\|^2 - \left\| z_{t+1} - z_{t+\frac{1}{2}} \right\|^2 \right). \tag{25}$$

We set $p = z^*$ to be a solution of the variational inequality (VI) problem in the above inequality. Note that

$$-2\eta \left\langle F(z_{t+\frac{1}{2}}), z_{t+\frac{1}{2}} - z^* \right\rangle = -2\eta \left\langle F(z_{t+\frac{1}{2}}) - F(z^*), z_{t+\frac{1}{2}} - z^* \right\rangle - 2\eta \left\langle F(z^*), z_{t+\frac{1}{2}} - z^* \right\rangle$$

$$\leq -2\eta \left\langle F(z^*), z_{t+\frac{1}{2}} - z^* \right\rangle \qquad (F \text{ is monotone})$$

$$= 2\eta \langle F(z^*), z_{t-1} - z^* \rangle - 4\eta \langle F(z^*), z_t - z^* \rangle \tag{26}$$

where the last equality holds since $z_{t+\frac{1}{2}} = 2z_t - z_{t-1}$. Also note that $\langle F(z^*), z_t - z^* \rangle \geq 0$ for all $t \geq 0$ since $z_t \in \mathcal{Z}$ and $z^*$ is a solution to (VI). Combing Inequality (25) and Inequality (26), telescoping the terms for $t = 1, 2, \cdots, T$, and dividing both sides by $1 - (1 + \sqrt{2})\eta L > 0$, we get

$$\sum_{t=1}^{T} \left\| z_{t+\frac{1}{2}} - z_t \right\|^2 \leq \frac{\|z_1 - z^*\|^2 + \|z_1 - z_{\frac{1}{2}}\|^2 + 2\eta \langle F(z^*), z_0 - z^* \rangle}{1 - (1 + \sqrt{2})\eta L}.$$

To get a cleaner constant that only relies on the starting point $z_0 = z_{\frac{1}{2}}$, we further simplify the three terms on the right-hand side. Note that since $\eta < \frac{1}{2L}$ and $z_1 = \Pi_{\mathcal{Z}}[z_0 - \eta F(z_0)]$, we have

$$\left\| z_1 - z_{\frac{1}{2}} \right\|^2 = \|z_1 - z_0\|^2 \leq \eta^2 \|F(z_0)\|^2 \leq \frac{4}{L^2} \|F(z_0)\|^2.$$

Thus we have

$$\|z_1 - z^*\|^2 \leq 2\|z_1 - z_0\|^2 + 2\|z_0 - z^*\|^2 \leq \frac{8}{L^2} \|F(z_0)\|^2 + 2\|z_0 - z^*\|^2.$$

Moreover,

$$\begin{aligned}
2\eta \langle F(z^*), z_0 - z^* \rangle &\leq 2\eta \|F(z^*)\| \|z_0 - z^*\| \\
&\leq 2\eta (\|F(z^*) - F(z_0)\| + \|F(z_0)\|) \|z_0 - z^*\| \quad (\|A\| \leq \|A - B\| + \|B\|) \\
&\leq 2\eta L \|z_0 - z^*\|^2 + 2\eta \|F(z_0)\| \|z_0 - z^*\| \\
&\leq \|z_0 - z^*\|^2 + \frac{1}{L} \|F(z_0)\| \|z_0 - z^*\| \quad (\eta < \tfrac{1}{2L}) \\
&\leq 2\|z_0 - z^*\|^2 + \frac{1}{L^2} \|F(z_0)\|^2. \quad (2ab \leq a^2 + b^2)
\end{aligned}$$

Thus

$$\|z_1 - z^*\|^2 + \left\| z_1 - z_{\frac{1}{2}} \right\|^2 + 2\eta \langle F(z^*), z_0 - z^* \rangle \leq \frac{13}{L^2} \|F(z_0)\|^2 + 4\|z_0 - z^*\|^2 = H^2.$$

This completes the proof. $\qquad\square$

**Lemma 7.** *In the same setup of Theorem 3, for any $T \geq 1$, we have*

$$\sum_{t=1}^{T} P_t \leq \lambda^2 H^2 L^2.$$

*Proof.* We first show an upper bound for $P_t$

$$\begin{aligned}
P_t &= \|F(z_t) + c_t\|^2 + \left\| F(z_t) - F(z_{t-\frac{1}{2}}) \right\|^2 \\
&= \left\| F(z_t) - F(z_{t-\frac{1}{2}}) + \frac{z_t - z_{t-1}}{\eta} \right\|^2 + \left\| F(z_t) - F(z_{t-\frac{1}{2}}) \right\|^2 \quad \text{(definition of } c_t \text{ (20))} \\
&\leq 3\left\| F(z_t) - F(z_{t-\frac{1}{2}}) \right\|^2 + \frac{2}{\eta^2} \|z_t - z_{t-1}\|^2 \quad (\|A + B\|^2 \leq 2\|A\|^2 + 2\|B\|^2) \\
&\leq 3L^2 \left\| z_t - z_{t-\frac{1}{2}} \right\|^2 + \frac{2}{\eta^2} \|z_t - z_{t-1}\|^2 \quad (F \text{ is } L\text{-Lipschitz}) \\
&= 3L^2 \left\| z_t - z_{t-1} + z_{t-1} - z_{t-\frac{1}{2}} \right\|^2 + \frac{2}{\eta^2} \|z_t - z_{t-1}\|^2 \\
&\leq 6L^2 \left\| z_{t-\frac{1}{2}} - z_{t-1} \right\|^2 + \left( \frac{2}{\eta^2} + 6L^2 \right) \|z_t - z_{t-1}\|^2 \quad (\|A + B\|^2 \leq 2\|A\|^2 + 2\|B\|^2) \\
&\leq \frac{2 + 6\eta^2 L^2}{\eta^2} \left( \left\| z_{t-\frac{1}{2}} - z_{t-1} \right\|^2 + \|z_t - z_{t-1}\|^2 \right).
\end{aligned}$$

Summing the above inequality of $t = 1, 2, \cdots T$, we get

$$
\begin{aligned}
\sum_{t=1}^{T} P_t &\leq \frac{2 + 6\eta^2 L^2}{\eta^2} \sum_{t=1}^{T} \left( \left\| z_{t-\frac{1}{2}} - z_{t-1} \right\|^2 + \| z_t - z_{t-1} \|^2 \right) \\
&= \frac{2 + 6\eta^2 L^2}{\eta^2} \left( \| z_1 - z_0 \|^2 + \sum_{t=1}^{T-1} \left( \left\| z_{t+\frac{1}{2}} - z_t \right\|^2 + \| z_{t+1} - z_t \|^2 \right) \right) \\
&\leq \frac{2 + 6\eta^2 L^2}{\eta^2} \left( \| z_1 - z_0 \|^2 + \frac{2H^2}{1 - (1 + \sqrt{2})\eta L} \right) \\
&\leq \frac{6(1 + 3\eta^2 L^2) H^2}{\eta^2 (1 - (1 + \sqrt{2})\eta L)}.
\end{aligned}
$$

The second last inequality holds by Lemma 6. The last inequality holds since $\| z_1 - z_0 \|^2 \leq \frac{4}{L^2} \| F(z_0) \|^2 \leq H^2$. Recall that $\lambda = \sqrt{\frac{6(1 + 3\eta^2 L^2)}{\eta^2 L^2 (1 - (1 + \sqrt{2})\eta L)}}$. This completes the proof. $\qquad\square$

### D.3 Proof of Theorem 3

Fix any $T \geq 1$. From Lemma 5, we know that the potential function $P_t$ is non-increasing for all $t \geq 1$. Lemma 7 guarantees that the sum of potential functions $\sum_{t=1}^{T} P_t$ is upper bounded by $\lambda^2 H^2 L^2$, where $\lambda^2 = \frac{6(1 + 3\eta^2 L^2)}{\eta^2 L^2 (1 - (1 + \sqrt{2})\eta L)}$. Combining the above, we can conclude that the potential function at the last iterate $P_T$ is upper bounded by $\frac{\lambda^2 H^2 L^2}{T}$. Since $P_T = \| F(z_T) + c_T \|^2 + \| F(z_T) - F(z_{T-\frac{1}{2}}) \|^2$, we obtain the last-iterate convergence rate $r_{F,Z}^{tan}(z_T)^2 \leq \| F(z_T) + c_T \|^2 \leq \frac{\lambda^2 H^2 L^2}{T}$.

The convergence rate on $\| F(z_T) + c_T \|^2$ implies a convergence rate on the gap function $\mathrm{GAP}_{Z,F,D}(z_T)$ by Lemma 1:

$$
\mathrm{GAP}_{\mathcal{Z},F,D}(z_T) \leq D \cdot \| F(z_T) + c_T \| \leq \frac{\lambda D H L}{\sqrt{T}}.
$$

## E Numerical Illustration

In this section, we conduct numerical experiments to illustrate and compare the performance of several algorithms: Reflected Gradient (RG), Extra Gradient (EG), Accelerated Reflected Gradient (ARG), and Fast Extra Gradient (FEG) (Lee & Kim, 2021a). Among them, ARG and FEG are accelerated algorithms while RG and EG are normal algorithms.

**Test Problem**   We use a classical example (Problem 1 in (Malitsky, 2015)) which is unconstrained and the operator $F(z) = Az$ where $A$ is an $n \times n$ matrix that

$$
A(i,j) = \begin{cases} 1, & j = n + 1 - i > i \\ -1, & j = n + 1 - i < i \\ 0, & \text{otherwise} \end{cases}
$$

Note that $F$ is 1-Lipschitz and its solution is the zero vector $\mathbf{0}$ when $n$ is even.

**Test Details**   We run experiments using Python 3.9 on jupyter-notebook, on MacBook Air (M1, 2020) running macOS 12.5.1. Time of execution is measured using the `time` package in Python. For all tests, we take initial point to be the all-one vector $z_0 = (1, \cdots, 1)$. We denote $\eta$ to be the step size and the termination criteria is the residual (operator norm) $\| F(z_t) \| \leq \varepsilon$. The code can be found in the Supplementary Material.

**Test Results**    The results for EG and RG are shown in Figure 1. With step size $\eta = 0.4$, EG is slower than RG. This is due to the fact that EG makes two gradient calls per iteration. Even with the optimized step size $\eta = 0.7$ which gives the best performance, EG is still slower than RG for this problem. Our results are consistent with numerical results on Mathematica by Malitsky (2015).

The results for FEG and ARG are shown in Figure 2. With step size $\eta = 0.5$, FEG is slower than ARG. With the optimized step size $\eta = 1$, FEG is a little faster than ARG. So for this problem, the performance of FEG and ARG are comparable. We also remark that for this particular problem, both ARG and FEG are slower than EG or RG. This does not contradict with our theoretical results on worst-case convergence rate. Simple algorithms like RG and EG can be faster than accelerated methods like ARG and FEG for particular problems. This also illustrates the importance of understanding simple algorithms like RG.

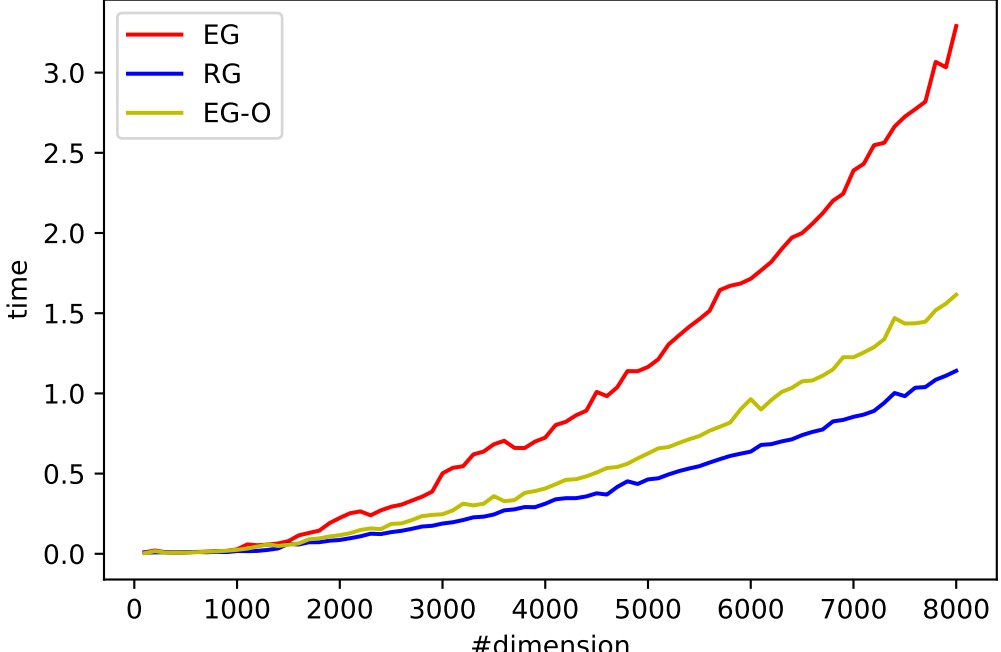

Figure 1: Results for EG and RG when $\varepsilon = 0.001$. The read line and blue line are EG and RG with step size $\eta = 0.4$. The yellow line is EG with (approximately) optimized step size $\eta = 0.7$. We remark that RG would diverge with $\eta = 0.7$.

## F   AUXILIARY PROPOSITIONS

**Proposition 3** (Two Identities). *Let* $(x_k)_{k\in[4]}$*,* $(y_k)_{k\in[4]}$*,* $x_0$*,* $u_2$ *and* $u_4$ *be arbitrary vectors in* $\mathbb{R}^n$*. Let* $k \geq 1$ *and* $q \in (0,1)$ *be two real numbers. If the following two equations holds:*

$$x_3 = x_2 - y_1 - u_2$$
$$x_4 = x_2 - y_3 - u_4$$

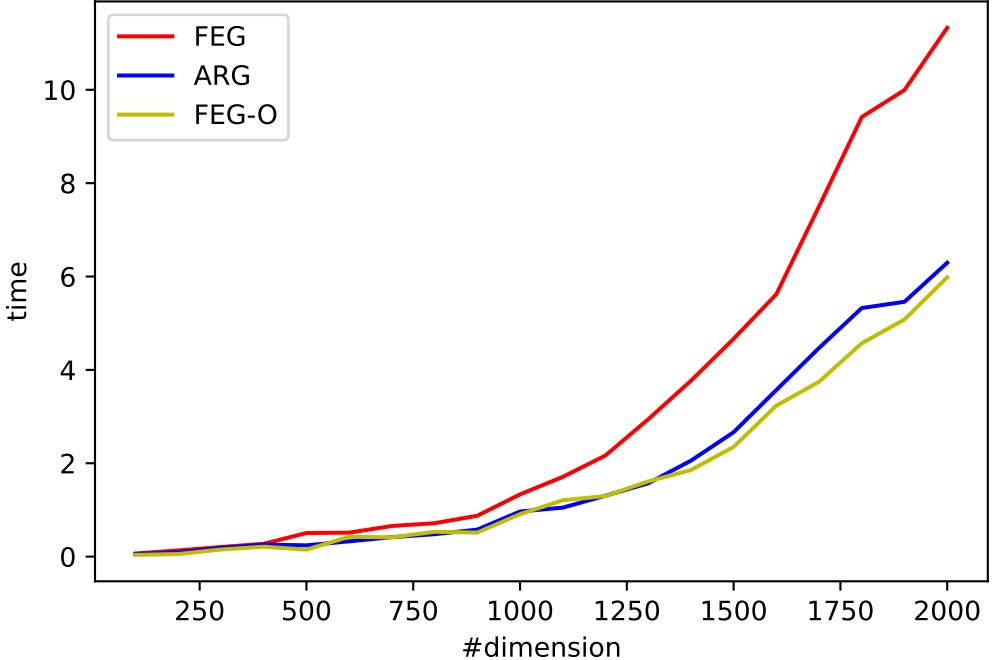

Figure 2: Results for FEG and ARG when $\varepsilon = 0.01$. The read line and blue line are FEG and ARG with step size $\eta = 0.5$. The yellow line is FEG with (approximately) optimized step size $\eta = 1$.

*then the following identity holds:*

$$
\begin{aligned}
&\|y_2 + u_2\|^2 + \|y_2 - y_1\|^2 - \|y_4 + u_4\|^2 - \|y_4 - y_3\|^2 \\
&- 2 \cdot \langle y_4 - y_2, x_4 - x_2 \rangle \\
&- 2 \cdot \left( \frac{1}{4} \cdot \|x_4 - x_3\|^2 - \|y_4 - y_3\|^2 \right) \\
&- 2 \cdot \langle u_4 - u_2, x_4 - x_2 \rangle \\
=& \left\| \frac{x_3 - x_4}{2} + y_1 - y_2 \right\|^2 + \left\| \frac{x_3 + x_4}{2} - x_2 + y_2 + u_2 \right\|^2
\end{aligned}
$$

*If the following two equations holds:*

$$
x_3 = x_2 - y_1 - u_2 + \frac{1}{k+1}(x_0 - x_2)
$$

$$
x_4 = x_2 - y_3 - u_4 + \frac{1}{k+1}(x_0 - x_2)
$$

*then the following identity holds:*

$$\frac{k(k+1)}{2}\left(\|y_2 + u_2\|^2 + \|y_2 - y_1\|^2\right) + k\langle y_2 + u_2, x_2 - x_0\rangle$$
$$- \frac{(k+1)(k+2)}{2}\left(\|y_4 + u_4\|^2 + \|y_4 - y_3\|^2\right) - (k+1)\langle y_4 + u_4, x_4 - x_0\rangle$$
$$- k(k+1) \cdot \langle y_4 + u_4 - y_2 - u_2, x_4 - x_2\rangle$$
$$- \frac{k(k+1)}{4q} \cdot \left\langle q \cdot \|x_4 - x_3\|^2 - \|y_4 - y_3\|^2\right\rangle$$
$$= \frac{k(k+1)}{4} \cdot \|u_4 - u_2 + y_1 - 2y_2 + y_3\|^2$$
$$+ \left(\frac{(1-4q)k - 4q}{4q}(k+1)\right) \cdot \|y_3 - y_4\|^2$$
$$+ (k+1) \cdot \langle y_3 - y_4, y_4 + u_4\rangle$$

*Proof.* We verify the two identities by MATLAB. The code is available at
https://github.com/weiqiangzheng1999/Single-Call. □

**Proposition 4** (([Cai et al., 2022a]))**.** *Let $\{a_k \in \mathbb{R}^+\}_{k \geq 2}$ be a sequence of real numbers. Let $C_1 \geq 0$ and $p \in (0, \frac{1}{3})$ be two real numbers. If the following condition holds for every $k \geq 2$,*

$$\frac{k^2}{4} \cdot a_k \leq C_1 + \frac{p}{1-p} \cdot \sum_{t=2}^{k-1} a_t, \tag{27}$$

*then for each $k \geq 2$ we have*

$$a_k \leq \frac{4 \cdot C_1}{1 - 3p} \cdot \frac{1}{k^2}. \tag{28}$$

*Proof.* We prove the statement by induction.

**Base Case: $k = 2$.** From Inequality (27), we have

$$\frac{2^2}{4} \cdot a_2 \leq C_1 \quad \Rightarrow \quad a_2 \leq C_1 \leq \frac{4 \cdot C_1}{1 - 3p} \cdot \frac{1}{2^2}.$$

Thus, Inequality (28) holds for $k = 2$.

**Inductive Step: for any $k \geq 3$.** Fix some $k \geq 3$ and assume that Inequality (28) holds for all $2 \leq t \leq k-1$. We slightly abuse notation and treat the summation in the form $\sum_{t=3}^{2}$ as 0. By Inequality (27), we have

$$\frac{k^2}{4} \cdot a_k \leq C_1 + \frac{p}{1-p} \cdot \sum_{t=2}^{k-1} a_t$$
$$\leq \frac{C_1}{1-p} + \frac{p}{1-p} \cdot \sum_{t=3}^{k-1} a_t \qquad\qquad (a_2 \leq C_1)$$
$$\leq \frac{C_1}{1-p} + \frac{4p \cdot C_1}{(1-p)(1-3p)} \cdot \sum_{t=3}^{k-1} \frac{1}{t^2} \qquad \text{(Induction assumption on Inequality (28))}$$
$$\leq \frac{C_1}{1-p} + \frac{2p \cdot C_1}{(1-p)(1-3p)} \qquad\qquad (\sum_{t=3}^{\infty} \frac{1}{t^2} = \frac{\pi^2}{6} - \frac{5}{4} \leq \frac{1}{2})$$
$$= \frac{C_1}{1-3p}.$$

This complete the inductive step. Therefore, for all $k \geq 2$, we have $a_k \leq \frac{4 \cdot C_1}{1-3p} \cdot \frac{1}{k^2}$. □

