# OpenReview forum: "Accelerated Single-Call Methods for Constrained Min-Max Optimization"
_ICLR.cc/2023/Conference — ICLR 2023 poster_

### Official Review · Reviewer_Sm3u · 2022-10-24

**Confidence:** 3
**Correctness:** 4
**Technical Novelty And Significance:** 2
**Empirical Novelty And Significance:** Not applicable
**Recommendation:** 3

**Clarity, Quality, Novelty And Reproducibility:**

Clarity: The problem setting is easy to understand.

Quality: The theoretical part establishes some new results and is clear. However, the single-call regime is not well-motivated and lacks numerical support.

Novelty: This paper studies a novel single-call nonconvex-nonconcave min-max problem.

Reproducibility: No numerical results are provided.

**Strength And Weaknesses:**

Strength:

1. The algorithms establish the first convergence guarantee for nonconvex-nonconvex min-max optimization problems under weak MVI and negative monotonicity regimes. These results are new.

Weaknesses:

1. This paper studies the single-call scenario that restricts algorithms to conduct one oracle call per iteration. This seems to be restrictive and lacks motivation. The classical extra-gradient method only requires making two calls to obtain the gradients. Decreasing two calls to a single call does not seem to have a significant practical impact. These differences are often hidden in the big-O $O(\cdot)$ notation with many other constants, such as the $H^2$ and $\eta^2$ terms specified in Theorem 1. Meanwhile, if we consider  $z = [x, y]$ as one vector, EG approaches only require one projection instead of two.  It would be beneficial if the authors could provide more motivation for studying single-call algorithms. For instance, is there a scenario in which only single-call algorithms can be applied?

2. In Theorem 1, the authors develop the convergence guarantees for weak MVI problems for a restrictive choice of $\rho \in ( - \frac{1}{12\sqrt{3}L}, 0]$. I wonder why it is necessary for the convergence claim. Is this choice of $\rho$ commonly accepted by EG methods for weak MVI problems? Similarly, Theorem 2 considers the case when $\rho \in ( -\frac{1}{100L}, 0 ]$, which also restricts its applicability. A comparison of choices of $\rho$ in existing works would be beneficial.

3. Theorem 3 establishes an $O(1/\sqrt{T})$ convergence guarantee for convex-concave problems, which is suboptimal for EG methods. In this case, it seems suboptimal to consider single-call approaches because it requires a larger amount of iterations than the classical EG methods.

4. This paper does not provide numerical experiments to test the practical performance of their algorithms. It would be beneficial if the authors could provide some numerical results to test their algorithm and compare the total required calls with classical EG approaches.

**Summary Of The Paper:**

This paper proposes three single-call algorithms for constrained min-max optimization and develops the following convergence guarantees. (i) The OG method achieves $O(1/\sqrt{T})$ convergence rate for weak MVI problems. (ii) The ARG method achieve the optimal $O(1/T)$ convergence rate in the negative comonotonicity regime. (iii) The RG method achieves an $O(1/\sqrt{T})$ last-iterate convergence guarantee for constrained convex-concave optimization. The weak MVI and negative comonotonicity regimes have been well studied.

**Summary Of The Review:**

This paper proposes a class of single-call algorithms for solving nonconvex-nonconcave min-max problems and establishes the theoretical convergence guarantees. However, the single-call regime is not well-motivated. Theorems 1 and 2 rely on some restrictive choices of $\rho$, which weakens their applicability. Theorem 3 provides a suboptimal convergence rate. No numerical results are provided to test the practical performance.

---

> ### Author Response · Authors · 2022-11-12
> **Response to Reviewer Sm3u**
>
> Thank you for you detailed and helpful comments! Below we first respond to your summary of the review and then the detailed comments on the weakness of the paper.
>
> *Summary of the review: "[...] However, the single-call regime is not well-motivated. Theorems 1 and 2 rely on some restrictive choices of $\rho$, which weakens their applicability. Theorem 3 provides a suboptimal convergence rate. No numerical results are provided to test the practical performance.*"
>
> We have included numerical experiments to illustrate the superior performance of single-call algorithms compared to two-call algorithms (See details in Appendix E and our response to 1 below).  Our ranges of $\rho$ for Theorem 1 and 2 are smaller than previous works by only a *constant factor* and the dependence on ${1\over L}$ is necessary (See our response to 2 below). Theorem 3 provides a tight last-iterate convergence rate for RG, and the two-call algorithm EG has the same last-iterate convergence rate (See our response to 3 below).
>
> 1. *"[...] Meanwhile, if we consider $z = [x,y]$ as one vector, EG approaches only require one projection instead of two. It would be beneficial if the authors could provide more motivation for studying single-call algorithms. For instance, is there a scenario in which only single-call algorithms can be applied?"*
>
> We do indeed consider $z=[x,y]$ in our paper (for min-max optimization problems). However, the update rule of EG is $z_{t+\frac{1}{2}} = \mathrm{proj}(z_t - F(z_t))$, $z_{t+1} = \mathrm{proj}(z_t - F(z_{t+\frac{1}{2}}))$, so it requires two oracle calls, i.e., $F(z_t)$ and $F(z_{t+{1\over 2}})$, and two projections in each iteration. We also test the practical performance of EG and the single-call algorithm RG by numerical experiments. Numerical results show that RG has better performance than EG even when EG uses the optimal step size. Our results are consistent with [Malitsky, 15] and shows that single-call algorithms can indeed be faster in practice especially in large-scale problems
> where the computation of gradient is costly.  Moreover, single-call algorithm is more natural in the online learning  setting. It is known that EG is not a no-regret algorithm while its single-call variant OG is no-regret. Therefore, our Theorem 1 provides the first convergence rate to Nash equilibrium for no-regret learning in non-monotone games that satisfy weak MVI.
>
> 2. *" [...] A comparison of choices of $\rho$ in existing works would be beneficial."*
>
> \noindent Thank you for pointing that out. Our choices of $\rho$ is worse than some of the previous works but only by a constant factor. Specifically, for the negatively comonotone setting, the best range of $\rho$ in previous works is $\rho \ge -\frac{1}{2L}$ and in our work is $\rho \ge -\frac{1}{100L}$. For the weak Minty setting, the best range of $\rho$ in previous works is $\rho \ge -\frac{1}{2L}$, and in our work is $\rho \ge -\frac{1}{12\sqrt{3}L}$. There is also evidence that the range of $\rho \ge -\frac{1}{2L}$ is necessary for convergence of an explicit algorithm [Pethick et al. 22]. Moreover, we remark that the dependence on $\frac{1}{L}$ can be viewed as normalization. Consider an $1$-Lipschitz operator $F$ that is $\rho$-comonotone for some $\rho<0$, i.e., $\langle F(z)- F(z'),z-z'\rangle\geq \rho\cdot ||F(z)-F(z')||^2$ for all $z,z'$. If we scale the operator by $L$, then the scaled operator $L\cdot F$ is $L$-Lipschitz and satisfies $\langle L\cdot F(z)-L\cdot F(z'),z-z'\rangle \geq {\rho \over L}\cdot || L\cdot F(z)-L\cdot F(z') ||^2$ for all $z,z'$, which means that the operator $L\cdot F$ is $\rho/L$-comonotone. We would also like to emphasize that compared to previous two-call algorithms, single-call single-resolvent algorithms would be more desirable for large-scale problems/online learning setting, as we discussed in responce to 1. Finding the optimal range of $\rho$ for single-call algorithms in the constrained setting is an interesting future direction.
>
> 3. *"Theorem 3 establishes an $O(\frac{1}{\sqrt{T}})$ convergence guarantee for convex-concave problems, which is suboptimal for EG methods. In this case, it seems suboptimal to consider single-call approaches because it requires a larger amount of iterations than the classical EG methods."*
>
> The $O({1\over T})$ convergence rate of EG is achieved by taking the average-iterate as the output. RG has the same $O({1\over T})$ average-iterate convergence rate w.r.t. the gap function [Hsieh et al.19]. However, in many settings, such as game theoretic ones, last-iterate convergence is more meaningful, and that is why [Hsieh et al. 2019] raises the last-iterate convergence rate of RG as an open problem. Note that for convex-concave problems, the last-iterate convergence rate of EG is also $O(\frac{1}{\sqrt{T}})$, which is the same as RG as shown in Theorem 3.

---

> > ### Author Response · Authors · 2022-11-12
> > **Response to Reviwer Sm3u (Part 2)**
> >
> > 4. *"This paper does not provide numerical experiments to test the practical performance of their algorithms. It would be beneficial if the authors could provide some numerical results to test their algorithm and compare the total required calls with classical EG approaches."*
> >
> > Thank you for the suggestion. We have included numerical results for RG, EG, ARG, and FEG in Appendix E in the revised paper. Our results show promising performances of single-call algorithms such as RG and ARG when compared to EG and FEG.
> >
> > **References**
> >
> > Malitsky, Yu. ”Projected reflected gradient methods for monotone variational inequalities.” SIAM
> > Journal on Optimization 25, no. 1 (2015): 502-520.
> >
> > Hsieh, Yu-Guan, Franck Iutzeler, Jérôme Malick, and Panayotis Mertikopoulos. "On the convergence of single-call stochastic extra-gradient methods." Advances in Neural Information Processing Systems 32 (2019).
> >
> > Pethick, Thomas, Panagiotis Patrinos, Olivier Fercoq, and Volkan Cevherå. "Escaping limit cycles: Global convergence for constrained nonconvex-nonconcave minimax problems." In International Conference on Learning Representations. 2022.

---

> > ### Comment · Reviewer_Sm3u · 2022-12-09
> > **Reply to the authors**
> >
> > I thank the authors for their detailed response and clarification on the single-call scheme, choice of $\rho$, theoretical results, and numerical experiments.  I agree with the authors that $\rho \geq -\frac{1}{2L}$ is standard, and the choice of $\rho$ in this paper differs by a constant ratio.  However, I would like to know what is the reason that causes such a difference in the choice of $\rho$. Could the authors briefly explain why they need this assumption and what is the intuition behind it? In addition, how would their algorithm behave when $\rho$ is chosen in the general setting $\rho\geq -\frac{1}{2L}$?

---

> > > ### Author Response · Authors · 2022-12-10
> > > **Response to Reviewer Sm3u**
> > >
> > > *Q: I would like to know what is the reason that causes such a difference in the choice of $\rho$. Could the authors briefly explain why they need this assumption and what is the intuition behind it? In addition, how would their algorithm behave when $\rho$ is chosen in the general setting $\rho \ge -\frac{1}{2L}$.*
> > >
> > > Thank you for the question. We suspect that our suboptimal range of $\rho$ is only an artifact of the analysis. We conducted numerical experiments on some instances from [Pethick et al. 2022]. The results show that Optimistic Gradient (OG) successfully escapes limit cycles and converges to  solutions of problems even when the corresponding $\rho$ is as small as $-\frac{1}{2L}$. Tightening our analysis to obtain the optimal range of $\rho$ is a very interesting question.
> > >
> > > It is worth mentioning that all previous theoretical results (e.g. EG+, CEG+) that allow a larger range of $\rho$ use **two-time scale** step sizes, while we only analyze the original OG, which uses a single step size. For example, the update rule of EG+ [Diakonikolas et al. 2021] is $z_{t+1/2} = z_t - \eta_1 F(z_t), z_{t+1} = z_t - \eta_2 F(z_{t+1/2})$ where $\eta_2 = \eta_1/2$. The only difference between EG and EG+ is that EG+ uses two different step sizes in the update rule, while EG uses the same step size. [Diakonikolas et al. 2021] shows that EG+ converges for $\rho \ge -\frac{1}{8L}$. More recently, [Pethick et al. 2022] proposes CEG+, a generalization of EG+ with a more sophisticated step size separation, and shows that CEG+ converges for $\rho \ge -\frac{1}{2L}$.  We analyze the original OG algorithm without step size separation. On the one hand, the lack of the additional flexibility (of using step size separation) makes the convergence analysis more challenging, and we only manage to guarantee convergence for a smaller range of $\rho$. On the other hand, OG is arguably simpler than EG+ and CEG+, and our result demonstrates the robustness of OG, as it retains the same convergence rate even in the nonconvex-nonconcave settings without any modification.
> > >
> > > **Reference**
> > >
> > > Pethick, Thomas, Panagiotis Patrinos, Olivier Fercoq, and Volkan Cevherå. "Escaping limit cycles: Global convergence for constrained nonconvex-nonconcave minimax problems." In International Conference on Learning Representations. 2022.
> > >
> > > Diakonikolas, Jelena, Constantinos Daskalakis, and Michael I. Jordan. "Efficient methods for structured nonconvex-nonconcave min-max optimization." International Conference on Artificial Intelligence and Statistics. PMLR, 2021.

---

### Official Review · Reviewer_qU6A · 2022-10-24

**Confidence:** 4
**Clarity, Quality, Novelty And Reproducibility:** The paper is written in a clear way a…
**Correctness:** 4
**Technical Novelty And Significance:** 4
**Empirical Novelty And Significance:** Not applicable
**Recommendation:** 8

**Strength And Weaknesses:**

Overall, I believe it is above bar for acceptance. The paper provides a clear but quite technical analysis (Proposition 2) and does a significant contribution to the field of single-call methods. Additionally, although the presented techniques might not be novel (see [3]), the authors propose the first single call single-resolvent method (ARG) of optimal convergence rate in the general structured (negatively comonotone) constraint nonconvex-nonconcave setting that has received important attention lately.

References
[1] Yang Cai, Argyris Oikonomou, and Weiqiang Zheng. Accelerated Algorithms for Monotone Inclusions and Constrained Nonconvex-Nonconcave Min-Max Optimization.
[2] Yu-Guan Hsieh, Franck Iutzeler, JÂ´erË†ome Malick, and Panayotis Mertikopoulos. On the convergence of single-call stochastic extra-gradient methods.
[3] Jelena Diakonikolas. Halpern iteration for near-optimal and parameter-free monotone inclusion and strong solutions to variational inequalities.

**Summary Of The Paper:**

The paper involves the study of constraint min-max optimization problems in the nonconvex-nonconcave setting with structure. Specifically, the more general framework of inclusion problems. Regarding the methods presented, they exclusively require single calls to the oracle and to the resolvent of an operator A; that is mainly the prominent goal of this work.
The two main contributions involve an extension of the Optimistic Gradient (OG) for the inclusion problems whose operators satisfied the weak MVI and a new accelerated Reflected Gradient (ARG) method for the same class of problems but for the case of negatively comonotone operators. The convergence rate achieved in the latter improves over [1] with respect to the number of oracle and projection calls, and is also optimal matching the lower bound of any first-order methods even for monotone inclusion problems. Lastly, the authors show that RG achieves a last-iterate convergence rate for convex-concave min max optimization, an open problem mentioned in [2].


References
[1] Yang Cai, Argyris Oikonomou, and Weiqiang Zheng. Accelerated Algorithms for Monotone Inclusions and Constrained Nonconvex-Nonconcave Min-Max Optimization.
[2] Yu-Guan Hsieh, Franck Iutzeler, JÂ´erË†ome Malick, and Panayotis Mertikopoulos. On the convergence of single-call stochastic extra-gradient methods.
[3] Jelena Diakonikolas. Halpern iteration for near-optimal and parameter-free monotone inclusion and strong solutions to variational inequalities.

Typos
	1	In the last expression of Theorem 1 there should \frac{2 \rho}{\eta}

**Summary Of The Review:**

The authors show that RG achieves a last-iterate convergence rate for convex-concave min max optimization, answering an open question asked in "On the convergence of single-call stochastic extra-gradient methods". This makes the paper above the bar for acceptance I feel.

---

> ### Author Response · Authors · 2022-11-12
> **Response to Reviewer qU6A**
>
> We thank the reviewer for the positive feedback.

---

### Official Review · Reviewer_GPXn · 2022-10-30

**Confidence:** 4
**Correctness:** 4
**Technical Novelty And Significance:** 2
**Empirical Novelty And Significance:** Not applicable
**Recommendation:** 5

**Clarity, Quality, Novelty And Reproducibility:**

### Missing related works
- the first contribution is as in [1] where a best-iterate rate is provided for a variant of OGDA (of the same order) for weak MVIs.
- a relevant work to the second contribution that achieves the same accelerated rate for OGDA-variant is [2] for monotone VIs.

These should be added in Table 1.

[1] Solving Nonconvex-Nonconcave Min-Max Problems exhibiting Weak Minty Solutions, Böhm, 2022, https://arxiv.org/abs/2201.12247

[2]  Fast OGDA in continuous and discrete time, Bot et al., 2022, https://arxiv.org/abs/2203.10947

### Questions
1. Since you are addressing the general case of general A, could you comment on some examples when A is not the normal cone for a closed domain? And are there some examples when the inverse computation is computationally inexpensive?
2. Could you comment on the range of convergent step sizes between OG and the two-projections methods that have the same rate on the setup of Thm 1? Did you explore trying to make Thm 1 parameter-free?
3. A similar technique of combining anchoring with EG was used by Lee & Kim, 2021. Could you comment on the technical differences (and challenges if applicable) when applying those techniques to OG?
4. Could you provide some examples in which cases one would use RG? Alternatively, could you provide some references to the relevant part in Sec. 5?
5. Could you provide some simulations of how ARG performs relative to accelerated variants of EG (or others) with guarantees for this setting?


### Writing / Minor / Typos
- Abstract: existing methods [...] require  (instead of *requires*)
- Abstract: missing quantifier, [...] in applications -> in some applications.
- Abstract: the abs. is wrongfully alluding that the first contribution is for the last iterate, and for standard OG, be specific that it is an OG-variant
- Abstract: only the third contribution specifies that the result is for the last iterate; be specific for the first two as well
- Abstract: specify the used measure for those rates
- page 2, last paragraph: operators satisfy the [...] -> operators that satisfy
- page 3: variationally stability -> variational stability
- page 3: either move the resolvent definition above or move the related works section after you define resolvent
- include full stops after the equations if the corresponding sentence ends after it.
- be specific in Theorem 1 and 2 to which assumption you refer when referencing Definition 1 since the latter has two different definitions
- Explain why you discuss Fact 1, instead of just listing it after the proof of the Theorem 1


**Strength And Weaknesses:**

**Strengths**.
- This paper focuses on some interesting and highly relevant VI problems and has several contributions.
- It is generally well-written, and easy to follow, with clear notation. I also like that the authors provide proof sketches when the full proof is deferred to the appendix.


**Weaknesses**
- Some of the presented aspects of the first two contributions exist in prior works (please see below). The third contribution does not exist if I am correct, but it is for the setting of a much smaller class of convex-concave zero-sum games.

- Missing motivation/relevance to focus on some of the considered methods and the little connection between the results and the considered setups and methods. In my opinion, it would have been clearer if the three results are separated and each is developed further with some more relevant questions such as parameter-free guarantees, stochastic case, etc, and discussing the insights from the theoretical results, e.g. better dependence on some constants relative to other methods, etc.

- Structure. This may be a consequence of the above, but it is often difficult to follow the flow in terms of the motivation of what follows. For example, the introduction mostly motivates the first two contributions. Please also discuss therein the difference between those two non-monotone setups, for completeness. Then, Sec. 5 separately discusses the motivation for Thm. 3 after the result is presented.

**Summary Of The Paper:**

The paper has several contributions:
1. it derives $(\mathcal{O}\frac{1}{\sqrt{T}})$ convergence rate of the best iterate of an extension of Optimistic gradient (OG). The setup is weak minty VIs (MVI)---which is a setting that includes non-monotone VIs as well; Sec. $3$.
2. proposes an accelerated version of Reflected Gradient (RG) by combining it with anchoring as in Halpern iteration--- named *ARG* for short---and shows that it achieves the optimal $\mathcal{O}(\frac{1}{T})$ last iterate rate. The setup here is negatively comonotone problems, which are larger than monotone but smaller class than weak MVIs.
3. shows that RG has $\mathcal{O}(\frac{1}{\sqrt{T}})$  last iterate rate for constrained convex-concave min-max optimization.


The first two results are in terms of the decreasing rate of a quantity called *tangent residual* which the authors show that it upper bounds the natural residual in Diakonikolas, 2020. The third one is with respect to the standard gap function and also the tangent residual.


**Summary Of The Review:**

This paper provides three contributions, among which the first two go beyond standard monotonicity, and thus are key to understanding game optimization.
Prior work showed similar results for the first two contributions. The third contribution focuses on a much smaller problem class and it is not clear to me why it is relevant that we consider that method given that other methods have the same rate and guarantees for larger problem classes, but I may be missing something.
It focuses on three different setups and three different methods, making its coherence and writing structure less easy to follow. The paper could be significantly improved by adding motivations and discussions for the presented results, as well as some further more practically relevant questions such as parameter-free guarantees and stochastic settings.

---

> ### Author Response · Authors · 2022-11-12
> **Responce to Reviewer GPXn (Part 1)**
>
> Thank you for you detailed and helpful comments! We have incorporated all your comments on minor issues in the revised paper. We also included numerical results that compare single-call algorithms such as RG and ARG with two-call algorithms such as EG and FEG in Appendix E in the revised paper. The changes are marked in red in the revised paper. Below we address your comments on the weaknesses and answer your questions.
>
> - *"Weakness: Some of the presented aspects of the first two contributions exist in prior works (please see below). The third contribution does not exist if I am correct, but it is for the setting of a much smaller class of convex-concave zero-sum games."*
>
> Our first two results provide the first single-call algorithms with the best known convergence rates for structured *constrained* nonconvex-nonconcave min-max optimization. Our results hold for any maximally monotone operator $A$, thus capture many additional settings (see our answer to Q1 below). In contrast, the mentioned recent works on single-call algorithms [Bohm, 22, Bot et al., 22] only consider the unconstrained setting where $A = 0$. Furthermore, [Bot et al., 22] only considers the monotone (convex-concave) setting, which is more restrictive than our setting. Our third contribution settles the open question by [Heish et al., 19] on *last-iterate* convergence rate of RG. RG is a particularly simple and basic algorithm, and we believe that understanding its last-iterate convergence rate is a worthwhile problem.
>
> 1. *Q1: "Since you are addressing the general case of general A, could you comment on some examples when A is not the normal cone for a closed domain? And are there some examples when the inverse computation is computationally inexpensive?"*
>
> When $A$ is the subdifferential operator of a lower semi-continuous, proper, and convex function $g$, i.e., $A=\partial g$, the inverse computation computes its resolvent $(I+\lambda \partial g)^{-1}$, which is also known as the *proximal operator* of $g$ denoted as $prox_{\lambda g}$.  The resolvent (proximal operator) $(I+\lambda \partial g)^{-1}$ is efficiently computable for the following popular choices of function $g$:  $\ell_1$-norm, $\ell_2$-norm, maxtrix norms, the log-barrier $-\sum_{i=1}^n \log (x_i)$, and more generally any quadratic or smooth functions.
> Moreover, many of them have closed-form expressions. For example, the proximal operator of the $\ell_1$-norm is the elementwise *soft-thresholding* operator $prox_{\lambda g}(v)[i]$$ = (v[i]-\lambda)^{+} - (-v[i]-\lambda)^{+}$.  We recommend the excellent survey by [Parikh and Boyd, 2014](Chapter 6 and 7) for a comprehensive review of proximal operators and their tractability. We have also added a discussion on the relation between resolvent operator and proximal operator and more examples in Appendix B.1 of the revised paper.
>
> 2. *Q2: "Could you comment on the range of convergent step sizes between OG and the two-projections methods that have the same rate on the setup of Thm 1? Did you explore trying to make Thm 1 parameter-free?"*
>
> The range of convergent step size for OG is smaller than that of EG. This is known for the monotone setting where EG allows $(0,1/L)$ step size and OG allows $(0, 1/2L)$ step size. The range of convergent step size for RG is also smaller than that of EG. Interestingly, we observe in our experiments that for large-scale problems, EG with the best step size can still be slower than RG, as the oracle call and projection can be highly costly in these settings. We haven't explored the possibility of making Thm 1 parameter-free. Making the algorithm parameter-free and work in the stochastic setting are very interesting future directions. Thanks for the suggestions.
>
> 3. *Q3: "A similar technique of combining anchoring with EG was used by Lee and Kim, 2021. Could you comment on the technical differences (and challenges if applicable) when applying those techniques to OG?"*
>
> We use anchoring to accelerate the RG algorithm. Compared to [Lee and Kim, 2021], who only considered the unconstrained setting, we consider the constrained setting and more generally any maximally monotone operator $A$ which allows us to capture many additional settings (see our answer to Q1). It turns out the constrained setting already makes the analysis more challenging. For example, the core of the analysis in [Lee and Kim, 2021] builds on a non-increasing potential function. This potential function is no longer non-increasing in the constrained setting. In fact, we did not find any non-increasing potential function in the constrained setting, making the analysis challenging. We overcame this challenge by constructing an approximately non-increasing potential, i.e., the potential only increases slowly between two consecutive iterates, and established the convergence rate with the help of this approximately non-increasing potential.

---

> ### Author Response · Authors · 2022-11-12
> **Response to Reviewer GPXn (Part 2)**
>
> 4. *Q4: "Could you provide some examples in which cases one would use RG? Alternatively, could you provide some references to the relevant part in Sec. 5?"*
>
> RG and its variants have been used in equilibrium computation problems [Guo et al, 2021; Alacaoglu et al, 2022]. We also remark that RG is more efficient than EG in some large-scale problems. The better performance of RG over the classical EG algorithm is illustrated through numerical results using Mathematica by Malitsky [2015]. The disadvantage of RG in comparison with two-call algorithms such as EG is its smaller range of step sizes. However, as observed in [Malitsky, 2015] *"For simple problems it really can matter, however, for huge-scale problems evaluation of F is much more important."* Malitsky conducted numerical experiments on a linear operator $F(z) = Az$ with a large $m\times m$ matrix $A$, where $m$ is either 2000 or 4000. Here are the two main findings
> : (1) when both algorithms are employed with the same step size, the performance of RG is much better than EG due to the extra costly projections and gradient calls made by EG. (2) even when EG is tuned with the *best* step size $\lambda=0.7$, EG is slower than RG with step size $\lambda = 0.4$ (RG dose not converge with $\lambda = 0.7$). We also conduct numerical experiments using Python (in Appendix E in the revised paper) and our result is consistent with [Malitsky, 2015].
>
> 5. *Q5: "Could you provide some simulations of how ARG performs relative to accelerated variants of EG (or others) with guarantees for this setting?"*
>
> We have included numerical results for ARG and FEG in Appendix E in the revised paper. The results show that when running with the same step size, ARG is faster than FEG. When FEG is tuned with the best step size, the performances of FEG and ARG are comparable.
>
> **References**
>
> Böhm, Axel. "Solving Nonconvex-Nonconcave Min-Max Problems exhibiting Weak Minty Solutions." arXiv preprint arXiv:2201.12247 (2022).
>
> Bot, Radu Ioan, Ernö Robert Csetnek, and Dang-Khoa Nguyen. "Fast OGDA in continuous and discrete time." arXiv preprint arXiv:2203.10947 (2022).
>
> Hsieh, Yu-Guan, Franck Iutzeler, Jérôme Malick, and Panayotis Mertikopoulos. "On the convergence of single-call stochastic extra-gradient methods." Advances in Neural Information Processing Systems 32 (2019).
>
> Parikh, Neal, and Stephen Boyd. "Proximal algorithms." Foundations and trends in Optimization 1.3 (2014): 127-239.
>
> Malitsky, Yu. "Projected reflected gradient methods for monotone variational inequalities." SIAM Journal on Optimization 25, no. 1 (2015): 502-520.
>
> Guo, Wenshuo, Michael I. Jordan, and Tianyi Lin. "A Variational Inequality Approach to Bayesian Regression Games." 2021 60th IEEE Conference on Decision and Control (CDC). IEEE, 2021.
>
> Alacaoglu, Ahmet, Luca Viano, Niao He, and Volkan Cevher. "A natural actor-critic framework for zero-sum Markov games." In International Conference on Machine Learning, pp. 307-366. PMLR, 2022.

---

### Decision · Program_Chairs · 2023-01-20

**Decision:**

Accept: poster

**Justification For Why Not Higher Score:**

The paper's technical strength and improvements over prior work are significant, but the assumptions and limitations of the analysis restrict the applicability of the results.

**Justification For Why Not Lower Score:**

The paper improves on prior works on several relevant dimensions and it introduces new analysis techniques that may lead to further progress.

**Metareview: Summary, Strengths And Weaknesses:**

The paper proposes new algorithms for structured non-convex non-concave min-max optimization that improve upon prior work by decreasing the number of gradient computations per iteration from two to one, by achieving last-iterate convergence, and by achieving improved convergence.

This paper generated significant discussion regarding the motivation for obtaining single-call methods and last iterate convergence, as well as the assumptions studied and the limitations of the analysis. After the discussion, the reviewers agreed that single call methods and last iterate convergence are meaningful contributions that are valuable to the community. The reviewers also agreed that the assumptions on the rho parameter are a significant weakness of the paper, but this limitation is shared to a large degree by the prior works as well.

Overall, the improvements over the prior work and the technical contributions are significant enough to warrant acceptance.

**Note From Pc:**

if the above contains the word "oral" or "spotlight" please see: "oral" presentation means -> notable-top-5% and "spotlight" means -> notable-top-25%. As stated in our emails, we are disassociating presentation type from AC recommendations

**Summary Of Ac-Reviewer Meeting:**

We discussed the motivation for obtaining single-call methods and last iterate convergence, as well as the assumptions studied and the limitations of the analysis. After the discussion, the reviewers agreed that single call methods and last iterate convergence are meaningful contributions that are valuable to the community. The reviewers also agreed that the assumptions on the rho parameter are a significant weakness of the paper, but this limitation is shared to a large degree by the prior works as well. We also discussed the technical contributions and the assumptions considered. The conclusion was that the technical contribution is strong and the assumptions are reasonable and mirror those in the prior work.